# Mapping patterns of thought onto brain activity during movie-watching

Raven Star Wallace[1]*, Bronte Mckeown[1], Ian Goodall-Halliwell[1], Louis Chitiz[1], Philippe Forest[2], Theodoros Karapanagiotidis[3], Bridget Mulholland[1], Adam Turnbull[4], Tamara Vanderwal[5], Samyogita Hardikar[6,7], Tirso RJ Gonzalez Alam[8], Boris C Bernhardt[9], Hao-Ting Wang[10], Will Strawson[3], Michael Milham[11], Ting Xu[11], Daniel S Margulies[12], Giulia L Poerio[3], Elizabeth Jefferies[13], Jeremy I Skipper[14], Jeffrey D Wammes[1], Robert Leech[15], Jonathan Smallwood[1]

[1]Department of Psychology, Queen's University, Kingston, Canada; [2]Mathematical and Electrical Engineering Department, IMT Atlantique, Brest, France; [3]School of Psychology, University of Sussex, Brighton, United Kingdom; [4]Department of Psychology, Stanford University, Stanford, United States; [5]Faculty of Medicine, University of British Columbia, Vancouver, Canada; [6]Department of Neurology, Max Planck Institute for Human Cognitive and Brain Sciences, Leipzig, Germany; [7]Max Planck School of Cognition, Leipzig, Germany; [8]School of Psychology and Sport Science, Bangor University, Gwynedd, United Kingdom; [9]Montreal Neurological Institute-Hospital, McGill University, Montreal, Canada; [10]Centre de Recherche de l'Institut Universitaire de Geriatrie de Montreal, Montreal, Canada; [11]Child Mind Institute, New York, United States; [12]Integrative Neuroscience and Cognition Center, University of Paris, Paris, France; [13]Division of Psychology & Language Sciences, University College London, London, United Kingdom; [14]Institute of Psychiatry, Psychology & Neuroscience, University College London, London, United Kingdom; [15]Department of Neuroimaging at the Institute of Psychiatry, Psychology and Neuroscience, King's College London, London, United Kingdom

*For correspondence: raven.wallace@queensu.ca

Competing interest: The authors declare that no competing interests exist.

## eLife Assessment

This study presents a **valuable** methodological advancement in quantifying thoughts over time. A novel multi-dimensional experience-sampling approach is presented, identifying data-driven patterns that the authors use to interrogate fMRI data collected during naturalistic movie-watching. The experimentation is inventive and the analyses carried out and results presented are **convincing**.

**Abstract** Movie-watching is a central aspect of our lives and an important paradigm for understanding the brain mechanisms behind cognition as it occurs in daily life. Contemporary views of ongoing thought argue that the ability to make sense of events in the 'here and now' depend on the neural processing of incoming sensory information by auditory and visual cortex, which are kept in check by systems in association cortex. However, we currently lack an understanding of how patterns of ongoing thoughts map onto the different brain systems when we watch a film, partly because methods of sampling experience disrupt the dynamics of brain activity and the experience of movie-watching. Our study established a novel method for mapping thought patterns onto the brain activity that occurs at different moments of a film, which does not disrupt the time course of brain activity or the movie-watching experience. We found moments when experience sampling

highlighted engagement with multi-sensory features of the film or highlighted thoughts with episodic features, regions of sensory cortex were more active and subsequent memory for events in the movie was better—on the other hand, periods of intrusive distraction emerged when activity in regions of association cortex within the frontoparietal system was reduced. These results highlight the critical role sensory systems play in the multi-modal experience of movie-watching and provide evidence for the role of association cortex in reducing distraction when we watch films.

## Introduction

A core goal of cognitive neuroscience is to understand how sensory input describing events in the external world is translated into the patterns of thoughts we experience in our lives. Complex naturalistic states, such as movie-watching, are important paradigms to understand this process because they allow cognition and brain dynamics to be understood in a situation that maps directly onto experiences in the real world (*Finn and Bandettini, 2021*; *Hasson et al., 2008b*; *Haxby et al., 2020*; *Vanderwal et al., 2019*; *Matusz, 2019*). Developments in cognitive neuroscience, leveraging state-of-the-art brain imaging techniques such as functional magnetic resonance imaging (fMRI), have established core features of neural patterns that emerge across participants during movie-watching tasks (*Hasson et al., 2008b*), highlighting their similarity across individuals (*Nastase et al., 2019*) and their links to memory for information in the films (*Hasson et al., 2008a*). However, it is more difficult to reliably map ongoing thought patterns in this context since experiential sampling, the gold-standard for tracking thought patterns (*Smallwood et al., 2021b*), has the potential to disrupt the natural unfolding of brain activity during movie-watching. The goal of our study was to minimize the disruptive impact of sampling ongoing experience by using a novel approach that allows us to explicitly link patterns of ongoing thought to brain activity during movie-watching at specific moments in a film.

Contemporary theories of ongoing thought suggest that a primary dimension to differentiate subjective experiences is the extent to which they depend on immediate sensory input (*Smallwood et al., 2021b*; *Smallwood, 2013a*). Cognitive states that are 'coupled' to events in the immediate environment are assumed to be linked to greater cortical processing of sensory input (*Smallwood et al., 2008a*), better task performance, and memory for events in narrative comprehension tasks like reading (*Smallwood and Andrews-Hanna, 2013*; *Smallwood et al., 2008b*; *Zhang et al., 2022*). In contrast, perceptually 'decoupled' states from sensory input, such as the experience of mind-wandering (*Smallwood and Schooler, 2006*; *Smallwood and Schooler, 2015*), provide an opportunity to pursue thoughts derived from memory (*Zhang et al., 2022*; *Medea et al., 2018*) but can be linked to compromised task performance and worse memory for events (*Schooler et al., 2011*). Moreover, in situations where comprehension is important, states of distraction are hypothesized to be linked to poor executive control (*Smallwood and Andrews-Hanna, 2013*; *Smallwood and Schooler, 2015*; *McVay and Kane, 2010*). Given that movie-watching provides a situation where dynamic changes in visual and auditory input drive a complex multi-sensory narrative, movie-watching provides an ecologically valid opportunity to understand how ongoing thought patterns map onto neural activation patterns in a naturalistic context. Recent work has shown that high-order regions, such as the ventromedial prefrontal cortex (vmPFC), are crucial in processing affective experiences during naturalistic stimuli, with distinct brain regions associated with different emotional expressions (*Chang et al., 2021*). These findings suggest that the brain undergoes continuous reorganization in naturalistic paradigms like movie-watching, and is sensitive to changes in psychological states, in this case affect. Our study aims to complement these approaches through the development of a novel experience sampling approach with which to understand how the changing patterns of brain activity that emerge during movie-watching relate to the different types of psychological experience that emerge in these moments of a film. In our study, we acquired experiential data in one group of participants while watching a movie clip and used these data to understand brain activity recorded in a second set of participants who watched the same clip and for whom no experiential data was recorded. This approach is similar to what is known as 'collaborative filtering' (*Chang et al., 2021*).

Consistent with the notion that movie-watching is a perceptually coupled state, recent work in cognitive neuroscience suggests an important role for primary systems, such as visual and auditory cortices (*Finn, 2021*). However, studies also hypothesize a role for regions of association cortex linked to higher order thought, such as the default mode network (DMN) or the frontoparietal network

(FPN; *Hasson et al., 2008b*; *Vanderwal et al., 2019*; *Yang et al., 2023*). For example, the DMN is hypothesized to be important in social cognition, episodic memory, and conceptual knowledge — all of which are likely important for understanding the narrative of the film (for a review of the broad role the DMN plays in cognition, see *Smallwood et al., 2021a*). However, the DMN has also been implicated in perceptually decoupled states, such as mind-wandering, that are likely antagonistic to movie-watching (*Zhang et al., 2022*; *Christoff et al., 2009*; *Konu et al., 2020*; *Zhang et al., 2019*).

Similarly, the FPN is important for multiple tasks, including those superficially different from movies, such as working memory maintenance, reflecting these networks' hypothesized role in goal maintenance (*D'Esposito and Postle, 2015*; *Chenot et al., 2021*). Contemporary views of ongoing thought argue that the FPN is likely important in suppressing distraction, including reductions in self-generated states like mind-wandering (*Vago and Zeidan, 2016*). Although studies have high-lighted the role of both primary sensory and higher-order systems in movie-watching (*Vanderwal et al., 2019*; *Rohr et al., 2018*), our lack of a formal understanding of the mapping between thought patterns when we watch films and associated patterns of brain activity means the specific role that different brain systems play in the experience of movie-watching remains largely a matter of specula-tion (*Demertzi et al., 2019*).

Our experiment was designed to better understand how the changing patterns of brain activity at different moments during a film map onto the ongoing thoughts accompanying them. Previous work has shown that shared conscious experiences can be linked to common neural codes, suggesting that when individuals engage in movie-watching, their brains may exhibit synchronized activity (*Chang et al., 2021*; *Naci et al., 2014*). Given that brain activity is synchronized across individuals, it could also be the case that there are shared thought patterns that also emerge (i.e. shared patterns of expe-rience) and if they do, these may also show links to common changes in brain activity. In our study, we used multi-dimensional experience sampling (mDES) to describe ongoing thought patterns during the movie-watching experience (*Smallwood et al., 2021b*). mDES is an experience sampling method that identifies different features of thought by probing participants about multiple dimensions of their experiences. mDES can provide a description of a person's thoughts, generating reliable thought patterns across laboratory cognitive tasks (*Smallwood et al., 2021a*; *Konu et al., 2021*; *Turnbull et al., 2021*) and in daily life (*Mckeown et al., 2021*; *Mulholland et al., 2023*), and is sensitive to accompanying changes in brain activity when reports are gained during scanning (*Konu et al., 2020*; *Turnbull et al., 2019a*). Studies that use mDES to describe experience ask participants to provide experiential reports by answering a set of questions about different features of their thought on a continuous scale from 1 (Not at all) to 10 (Completely; *Konu et al., 2020*; *Konu et al., 2021*; *Turnbull et al., 2021*; *Mckeown et al., 2021*; *Mulholland et al., 2023*; *Turnbull et al., 2019a*; *Ho et al., 2020*; *Karapanagiotidis et al., 2017*; *Mckeown et al., 2023*; *Vatansever et al., 2020*; *Wang et al., 2018*). Each question describes a different feature of experience, such as if their thoughts are oriented in the future or the past, about oneself or other people, deliberate or intrusive in nature, and more (see Methods for a full list of questions used in the current study).

One challenge that arises when attempting to map the dynamics of thought onto brain activity during movie-watching is accounting for the inherently disruptive nature of experience sampling: to measure experience with sufficient frequency to map experiential reports during movies would inher-ently disrupt the natural processes of the brain and alter the viewer's experience (for example, by pausing the film at a moment of suspense). Therefore, if we periodically interrupt viewers to acquire a description of their thoughts while recording brain activity, this could impact capturing important dynamic features of the brain. On the other hand, if we measured fMRI activity continuously over movie-watching (as is usually the case), we would lack the capacity to directly relate brain signals to the corresponding experiential states. Thus, to overcome this obstacle, we developed a novel meth-odological approach using two independent samples of participants. In the current study, one set of 120 participants was probed with mDES five times across the three ten-minute movie clips (11 min total, no sampling in the first minute). We used a jittered sampling technique where probes were delivered at different intervals across the film for different people depending on the condition they were assigned. Probe orders were also counterbalanced to minimize the systematic impact of prior and later probes at any given sampling moment. We used these data to construct a precise descrip-tion of the dynamics of experience for every 15 s of three 10-min movie clips. These data were then combined with fMRI data from a different sample of 44 participants who had already watched these

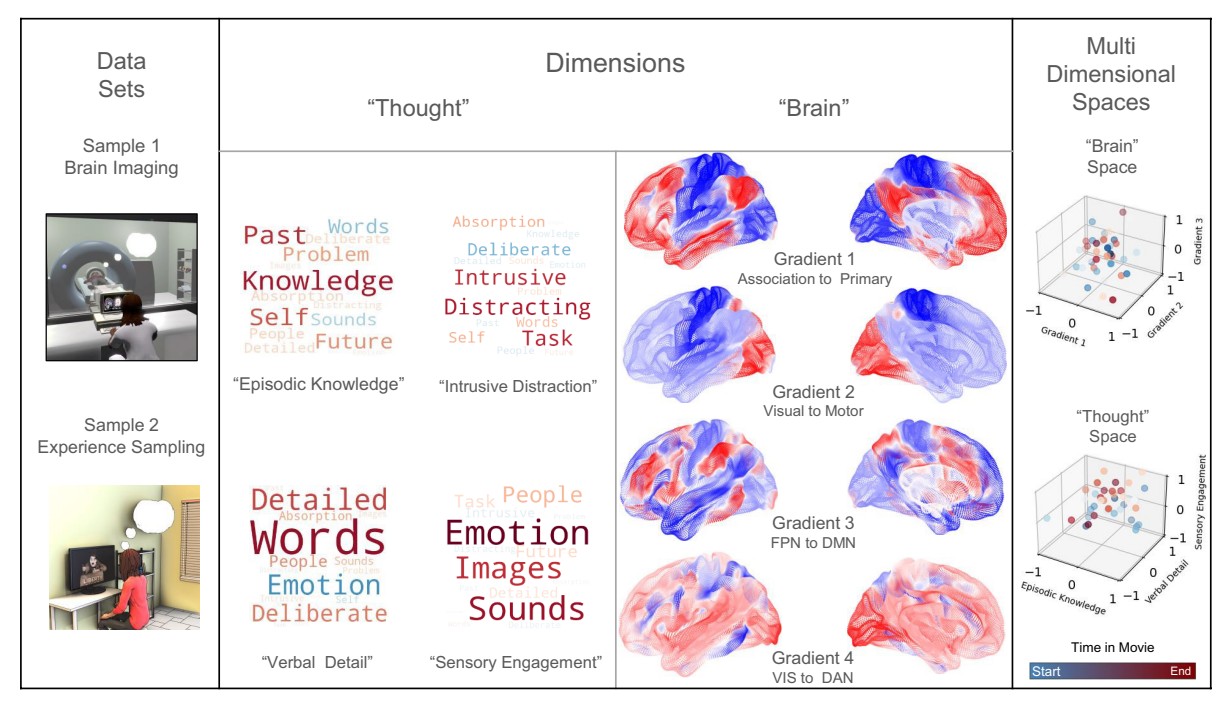

**Figure 1.** Using fMRI data and experience sampling data to map ongoing thought patterns onto brain activity during movie-watching. *Left to Right -* One sample of participants was scanned while watching movies (Sample 1), and a different set of participants responded to experience sampling probes (Sample 2) while watching the same movies in the laboratory. Decomposition of mDES data into low-dimension experiential patterns using principal component analysis (PCA) produced a set of dimensions that describe experience during movie-watching (a 'thought space' within which the dynamics of the movie-watching experience unfold). Word clouds illustrate how the experience sampling questions map onto each dimension that describes this space. In these word clouds, the font size describes their importance (bigger = more important), and the colour describes their polarity (red = positive, blue = negative). Similarly, we created a brain space to describe the movie-watching experience by comparing each moment in the film to validated dimensions of brain variation. For this purpose, we used the dimensions defined from the resting states of the HCP conducted by Margulies (***Margulies et al., 2016***) (often referred to as gradients): Gradient 1 (Association to Primary cortex), Gradient 2 (Visual to Motor cortex), Gradient 3 (Frontoparietal to Default Mode Networks), and Gradient 4 (Dorsal Attention Network (DAN)/Visual to Default Mode Networks) of brain variation dimensions illustrated by colour to map activity in state space analysis (purple = low, yellow = high) (not shown: Gradient 5 Lateral Default Mode to Primary sensory cortex) (***Margulies et al., 2016***). Two 3D scatter plots illustrating two examples from our data of how the movie-watching can be seen as two complimentary trajectories through a 'Brain Space' (focusing on Gradients 1, 2, and 3, shown at the top) and a 'Thought Space' (focusing on 'Episodic Knowledge', 'Verbal Detail', and 'Sensory Engagement', shown at the bottom). The cooler (blue) points occur earlier in the movie clip and the warmer (red) points occur later.

The online version of this article includes the following figure supplement(s) for figure 1:

**Figure supplement 1.** Scree plot of mDES thought data.

clips without experience sampling (***Aliko et al., 2020***). By combining data from two different groups of participants, our method allows us to describe the time series of different experiential states (as defined by mDES) and relate these to the time series of brain activity in another set of participants who watched the same films with no interruptions. In this way, our study set out to explicitly understand how the patterns of thoughts that dominate different moments in a film in one group of participants relate to the brain activity at these time points in a second set of participants and, therefore, better understand the contribution of different neural systems to the movie-watching experience.

## Results
### Analytic goal

The goal of our study, therefore, was to understand the association between patterns of brain activity over time during movie clips in one group of participants and the patterns of thought that participants reported at the corresponding moment in a different set of participants (see *Figure 1*). This can be conceptualized as identifying the mapping between two multi-dimensional spaces, one reflecting

the time series of brain activity and the other describing the time series of ongoing experience (see *Figure 1* right-hand panel). In our study, we selected three 11 mine clips from movies (*Citizenfour*, *Little Miss Sunshine* and *500 Days of Summer*) for which recordings of brain data in fMRI already existed (n=44) (*Aliko et al., 2020*; *Figure 1*, Sample 1). A second set of participants (n=120) viewed the same movie clips, providing intermittent reports on their thought patterns using mDES (*Figure 1*, Sample 2). Our goal was to understand the mapping between the patterns of brain activity at each moment of the film and the reports of ongoing thought recorded at the same point in the movies. We first applied Principal Components Analysis (PCA) to the mDES data to reduce these data to a set of four simple dimensions that explained the reported thought pattern. These are represented as word clouds in *Figure 1*. We performed two analyses to understand the associations between the reported thought patterns and brain activity at each point in the film. Our first analysis computed the mean time series of experience for each of the four thought pattern components (averaged across participants in Sample 2) and used this as a regressor of interest in a model predicting brain activity recorded from each participant from Sample 1. We refer to this as a *voxel-space* analysis, and it allowed us to perform a whole-brain search of the mapping between activity in each region to each dimension of ongoing thought. In our second analysis, we projected the grand mean of brain activity for each volume of each film against the first five dimensions of brain activity from a decomposition of the Human Connectome Project (HCP) resting state date to form a 5D 'brain space' that describes the trajectory of the brain during each movie (*Figure 1*, *note* only the first four dimensions are shown; *Margulies et al., 2016*). We used the results of this analysis to produce coordinates for each TR of each movie, which were used as explanatory variables in a linear mixed model (LMM) in which the location of each mDES probe in the 'thought space' described by the PCA dimensions were the dependent variables. We refer to this second analysis as a *state-space* analysis (see *Karapanagiotidis et al., 2017*; *Mckeown et al., 2023*; *Turnbull et al., 2020* for prior examples of this approach).

## Generation of the thought space

The first step in our analysis was to decompose the mDES data using PCA to produce the dimensions that comprise the 'thought space' used for our subsequent analyses (*Figure 1*, see Methods). Based on the scree plot (see *Figure 1—figure supplement 1*), the data best fit a four-component solution, and the resulting components are displayed as word clouds (see Methods for further details). In these word clouds, items with similar colours are related, and the font size indicates their importance. Component 1 contributed 26.1% of the variance explained and loaded positively on terms 'past', 'self', and 'knowledge', and negatively on 'words' and 'sounds', and is referred to as 'Episodic Knowledge'. Component 2 explained 10.5% of the variance and loaded positively on the items 'intrusive' and 'distracting' and negatively on 'deliberateness', and is referred to as 'Intrusive Distraction'. Component 3 loaded positively on 'words', 'detail', and 'deliberateness', explaining 7.7% of the variance, and is called 'Verbal Detail'. Finally, Component 4 contributed 6.8% of the explained variance, loaded positively on 'emotion', 'images', 'sounds', and 'people', and was named 'Sensory Engagement'. See *Supplementary file 1a* for a description of the mDES questionnaire and *Supplementary file 1b* for the percentage of variance explained by principal components overall and in each movie.

## Split-half reliability results

A bootstrapped split-half reliability analysis was conducted to confirm that the four-component solution provided a reasonable description of our data. This analysis repeatedly divided the mDES data into two random samples and evaluated the correlation between the two halves' components. The reliability analysis supported that the four-component solution was reproducible because it had a strong homologue similarity score (r=0.96, 95% CI [0.93, 1.00]; see Methods for further details).

## Variation in thought patterns

Next, we examined how these dimensions describe experience within each movie (see *Figure 2*). We performed four linear mixed models (LMM), one for each thought component ('Episodic Knowledge', 'Intrusive Distraction', 'Verbal Detail', and 'Sensory Engagement'), in which the movie was the explanatory variable of interest, and participants were included as a random effect. The significance threshold was adjusted using the False Discovery Rate (FDR) to control for family-wise error (FWE) within the model (controlling for the three movies). The four-model analyses found significant

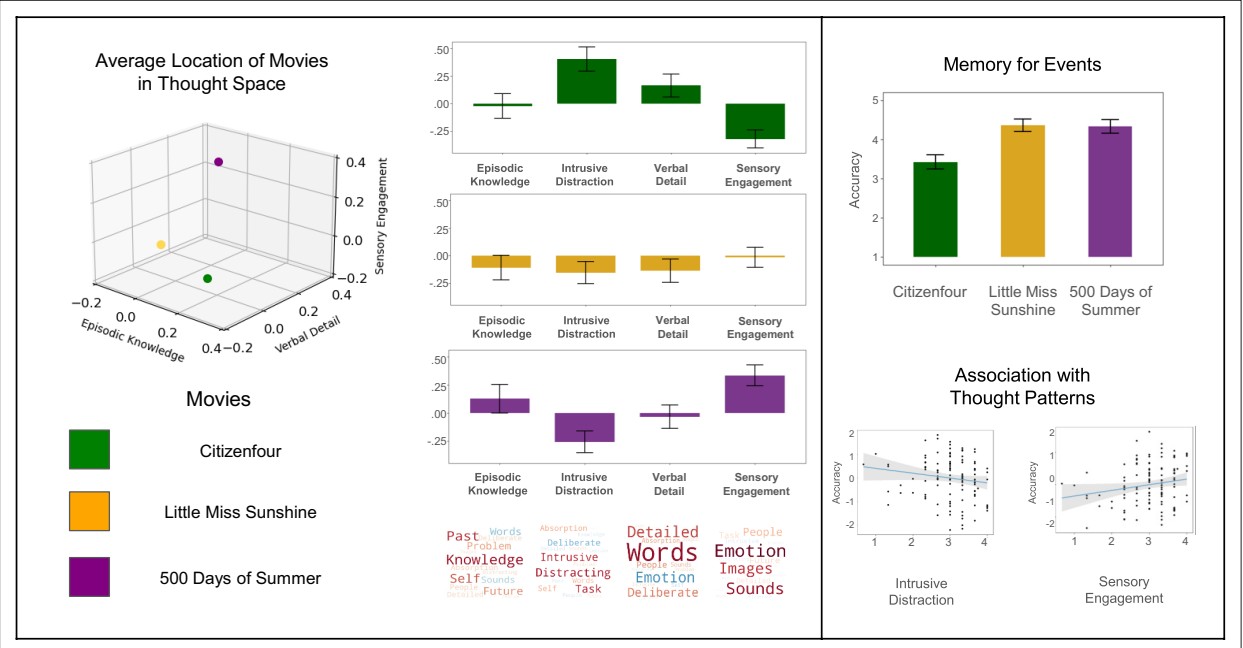

**Figure 2.** The relationship between how patterns differ across movie clips and relate to comprehension. *Left to Right* – The 3D scatterplot shows the average location of each film on three of the four PCA dimensions, 'Episodic Knowledge', 'Verbal Detail', and 'Sensory Engagement'. The bar graphs show the average loading on each dimension, with the error bars showing the 95% Confidence Interval. The plots on the right illustrate the relationship between the mDES dimensions and memory for information in the film. The top barplot shows the average comprehension score on each film with 95% Confidence Intervals error bars. The scatter plots below show the association between mDES components and comprehension. The scatter plot on the left shows the negative linear relationship between the 'Intrusive Distraction' thought and memory. The plot on the right shows a positive association with 'Sensory Engagement'. The blue line represents the best-fit line, and the shaded area shows the 95% Confidence Intervals.

differences in overall thought pattern scores across the three movies, including reported thoughts resembling 'Episodic Knowledge', $F(2, 2015.3)=5.41$, p=0.005, $\eta 2=0.01$, 'Intrusive Distraction', $F(2, 2015.3)=77.84$, p<0.001, $\eta 2=0.07$, 'Verbal Detail', $F(2, 2015.4)=13.90$, p<0.001, $\eta 2=0.01$, and 'Sensory Engagement', $F(2, 2015.7)=82.69$, p<0.001. $\eta 2=0.08$. This suggests that within each model, there was a significantly different score for the reported thought pattern in at least one of the movies. Post-hoc pairwise comparisons using the least-squares means (lsmeans) were conducted for each model to investigate how thought component scores differ in each movie, adjusting significance thresholds using the Tukey method to control for FWE within the model. The first model suggests patterns of responses in *Little Miss Sunshine* showed less similarity to 'Episodic Knowledge' (*M*=–0.12, SE = 0.10) than did patterns of thoughts reported in *500 Days of Summer* (*M*=0.11, SE = 0.10), $t(2016)=-3.27$, p=0.003. However, there were no significant differences in 'Episodic Knowledge' thoughts reported during *Citizenfour* (*M*=–0.02, SE = 0.10) compared to *Little Miss Sunshine*, $t(2015)=1.31$, p=0.392, or *500 Days of Summer*, $t(2016)=-1.97$, p=0.121. The second model identified self-reported thoughts that were more similar to the pattern of 'Intrusive Distraction' during *Citizenfour* (*M*=0.41, SE = 0.09) than during *Little Miss Sunshine* (*M*=–0.15, SE = 0.09), $t(2015)=9.66$, p<0.001, or during *500 Days of Summer* (*M*=–0.27, SE = 0.09), $t(2015)=11.66$, p<0.001. There was no difference in how similar reported thoughts scores were to 'Intrusive Distraction' between *Little Miss Sunshine* and *500 Days of Summer*, $t(2016)=2.03$, p=0.106. Model three found self-reported thoughts resemble patterns of 'Verbal Detail' more for *Citizenfour* (*M*=0.17, SE = 0.09) than for *Little Miss Sunshine* (*M*=–0.14, SE = 0.09), $t(2015)=5.14$, p<0.001, or *500 Days of Summer* (*M*=–0.04, SE = 0.09), $t(2016)=3.58$, p=0.001. Again, there were no significant differences in reported 'Verbal Detail' scores between *Little Miss Sunshine* and *500 Days of Summer*, $t(2016)=-1.54$, p=0.271. Lastly, model four found reported thoughts during *Citizenfour* resembled patterns of 'Sensory Engagement' (*M*=–0.32, SE = 0.07) significantly less than for either *Little Miss Sunshine* (*M*=–0.01, SE = 0.07), $t(2015)=-6.07$, p<0.001, or *500 Days of Summer* (*M*=0.33, SE = 0.07), $t(2016)=-12.85$, p<0.001. Additionally, reported thoughts during *Little Miss Sunshine* resembled patterns of 'Sensory Engagement' less than

**Table 1.** ANOVA across sampling bins of each Movie of each Thought Component score.

*Little Miss Sunshine*

|  |  | Df | Sum Sq | Mean Sq | *F*-value | p-value |
|---|---|---|---|---|---|---|
| PCA_1 | Sampling bin | 1.00 | 24.30 | 24.29 | 10.80 | 0.001 ** |
|  | Residuals | 712.00 | 1601.80 | 2.25 |  |  |
| PCA_2 | Sampling bin | 1.00 | 3.60 | 3.64 | 1.97 | 0.161 |
|  | Residuals | 712.00 | 1219.50 | 1.85 |  |  |
| PCA_3 | Sampling bin | 1.00 | 63.50 | 63.54 | 31.79 | <0.001 *** |
|  | Residuals | 712.00 | 1423.00 | 2.00 |  |  |
| PCA_4 | Sampling bin | 1.00 | 5.10 | 5.06 | 3.43 | 0.064 |
|  | Residuals | 712.00 | 1048.20 | 1.47 |  |  |

*Citizenfour*

|  |  | Df | Sum Sq | Mean Sq | *F*-value | p-value |
|---|---|---|---|---|---|---|
| PCA_1 | Sampling bin | 1.00 | 12.00 | 12.01 | 5.23 | 0.023 * |
|  | Residuals | 712.00 | 1637.00 | 2.30 |  |  |
| PCA_2 | Sampling bin | 1.00 | 2.00 | 1.95 | 0.87 | 0.350 |
|  | Residuals | 712.00 | 1593.00 | 2.24 |  |  |
| PCA_3 | Sampling bin | 1.00 | 0.10 | 0.07 | 0.04 | 0.847 |
|  | Residuals | 712.00 | 1425.30 | 2.00 |  |  |
| PCA_4 | Sampling bin | 1.00 | 7.40 | 7.40 | 6.22 | 0.013 * |
|  | Residuals | 712.00 | 847.80 | 1.19 |  |  |

*500 Days of Summer*

|  |  | Df | Sum Sq | Mean Sq | *F*-value | p-value |
|---|---|---|---|---|---|---|
| PCA_1 | Sampling bin | 1.00 | 7.30 | 7.34 | 2.51 | 0.114 |
|  | Residuals | 706.00 | 2068.50 | 2.93 |  |  |
| PCA_2 | Sampling bin | 1.00 | 0.20 | 0.22 | 0.13 | 0.719 |
|  | Residuals | 706.00 | 1219.10 | 1.73 |  |  |
| PCA_3 | Sampling bin | 1.00 | 7.80 | 7.85 | 3.86 | 0.049 * |
|  | Residuals | 712.00 | 1425.30 | 2.00 |  |  |
| PCA_4 | Sampling bin | 1.00 | 114.20 | 114.15 | 80.41 | <0.001 *** |
|  | Residuals | 706.00 | 1002.30 | 1.42 |  |  |

Note. Results of each ANOVA test assessing if each PCA thought component score differs across each of the 15 s sampling bins (Sampling bin) for the three movies Little *Miss Sunshine, Citizenfour* and *500 Days of Summer*. The table consists of the degrees of freedom (Df), sum of squares (Sum Sq), mean squares (Mean Sq), F-value, and p-value for each component and movie. Significant p-value (p<0.05) indicates a significant difference in the respective PCA component score across the sampling bins.

reported thoughts during *500 Days of Summer*, $t(2016)=-6.81$, p<0.001. The results are presented visually in *Figure 2*, and further details of the LMM are presented in *Supplementary file 1c*.

## Time series of experience across movie clips

Next, we examined how each pattern of thought changes across each movie clip. For this analysis, we conducted separate ANOVAs for each film clip for the four components (see *Table 1* and *Figure 3*). Clear dynamic changes were observed in several components for different films. First, there was a significant change in 'Episodic Social Cognition' scores across *Little Miss Sunshine*, $F(1, 712)=10.80$, p=0.001, $\eta 2=0.03$, and *Citizenfour*, $F(1, 712)=5.23$, p=0.023, $\eta 2=0.02$. There was also a significant

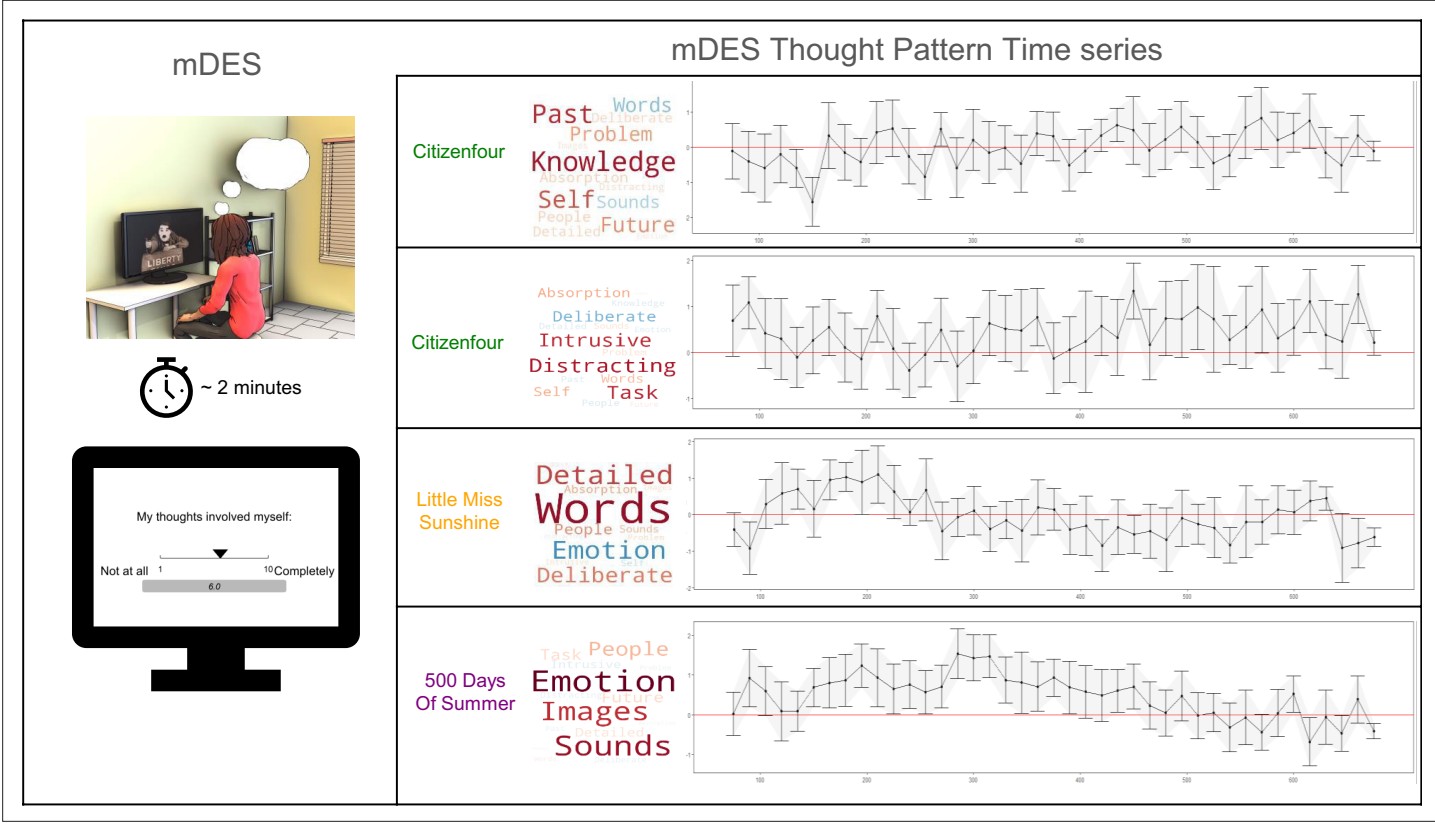

**Figure 3.** The application of multi-dimensional experience sampling (mDES) method and relevant time series produced from decomposed mDES thought patterns. *Left to Right* – The first panel illustrates the mDES method in the laboratory to demonstrate how participants respond to the sixteen items about their thoughts while watching the film on the laboratory computers. The plots on the right summarize the average thought pattern score at each 15 s sampling window across the three movies. The first time series plot illustrates the trajectory of the 'Episodic Knowledge' across *Little Miss Sunshine*, followed by the time course of 'Verbal Detail' also across *Little Miss Sunshine*, with distinct peaks in scores within the 150–250 s range and particularly low scores between the 400 and 500 s interval. The third plot demonstrates the relatively low and negative scores on 'Sensory Engagement' across *Citizenfour*. Lastly, the final plot highlights the relatively high scores on 'Sensory Engagement' throughout *500 Days of Summer*, especially across the 150–400 s interval.

The online version of this article includes the following figure supplement(s) for figure 3:

**Figure supplement 1.** Probe order matrix for Sample 2.

change in 'Verbal Detail' scores across *Little Miss Sunshine*, $F(1, 712)=31.79$, $p<0.001$, $\eta 2=0.09$. Lastly, there were significant changes in 'Sensory Engagement' scores for both *Citizenfour*, $F(1, 712)=6.22$, $p=0.013$, $\eta 2=0.02$, and *500 Days of Summer*, $F(1, 706)=80.41$, $p<0.001$, $\eta 2=0.18$. These time series are plotted in *Figure 3* and highlight how mDES can capture the dynamics of different types of experience across the three movie clips. Moreover, in several of these time series plots, it is clear that reported thought patterns extend beyond adjacent time periods (e.g. scores above zero between time periods 150–400 for Sensory Engagement in *500 Days of Summer* and for time periods between 175 and 225 for Verbal Detail in *Little Miss Sunshine*). It is important to note that no participant completed experience sampling reports during adjacent sampling points (see *Figure 3—figure supplement 1*), so the length of these intervals indicates agreement in how specific scenes within a film were experienced and conserved across different individuals. Notably, the component with the least evidence for temporal dynamics was 'Intrusive Distraction'.

## Comprehension

Next, we examined how the thought patterns relate to the participants' memory of information from the movies (*Figure 2*). Participants answered four comprehension questions for each film (12 total) related to relevant information in the clip they just watched (see *Supplementary file 1d* for the comprehension questionnaire). We performed an LMM for which the movies, each thought pattern,

and their interaction were explanatory variables of interest. Comprehension score was the dependent variable, and participant was included as a random effect. FDR was used to control for FWE, consisting of nine comparisons. The analysis revealed three significant main effects and a significant interaction. First, there was a significant main effect of movie on memory, $F(2, 254.12)=49.33$, p<0.001, $\eta 2=0.28$. Post-hoc pairwise comparisons using the lsmeans were conducted to investigate the effect of memory performance across the different films. Significance thresholds for the post-hoc comparisons were adjusted using the Tukey method to control FWE within the model. Comprehension scores were significantly lower for questions related to information in *Citizenfour* (*M*=2.42, SE = 0.09) compared to *Little Miss Sunshine* (*M*=3.35, SE = 0.08), $t(249) = -9.16$, p<0.001, as well as *500 Days of Summer* (*M*=3.33, SE = 0.08), $t(273) = -8.33$, p<0.001. Notably, there was no significant difference in comprehension performance between *Little Miss Sunshine* and *500 Days of Summer*, $t(242) = -0.18$, p=0.982. There were also two significant main effects of thought patterns — 'Intrusive Distraction' was significantly associated with worse comprehension across the three movies, $F(1, 324.41)=9.27$, p=0.011, $\eta 2=0.03$, whereas 'Sensory Engagement' was associated with better overall comprehension, $F(1, 341.44)=8.30$, p=0.013, $\eta 2=0.02$. Finally, there was a significant movie by thought pattern interaction for 'Episodic Knowledge', $F(2, 268.96)=4.46$, p=0.028, $\eta 2=0.03$. To follow up on this significant interaction, post-hoc simple slopes analysis was performed to assess the effect of 'Episodic Knowledge' on comprehension performance across each movie, using FDR to control for multiple comparisons. The analysis found moments when patterns of thought were more similar to 'Episodic Knowledge' were associated with significantly better comprehension performance for information in *500 Days of Summer*, $t(319.83)=2.54$, p=0.030. The interaction predicted negative comprehension performance for information in *Citizenfour*, $t(317.55)=-1.39$, *b*=-0.09, SE = 0.06, p=0.240, but positive comprehension performance for information in *Little Miss Sunshine*, $t(321.85)=0.93$, *b*=0.07, SE = 0.07, p=0.350, although neither of these relationships was statistically significant. To see the complete model output and the pairwise comparisons, see *Supplementary file 1e* One important implication of these results suggests mDES is sensitive to objective indicators of movie-watching experience because they indicate that individuals for whom self-reported experience shows less evidence of the pattern of 'Intrusive Distraction' tended to encode features of the movie more accurately and, therefore, performed better on the comprehension test.

## Brain – thought mappings: voxel-space analysis

Having established the dimensions that characterize the mDES data, how they organize experience in each movie, their variation over time, and their associations to memory, we then examined how these dimensions of experience relate to the brain activity at each moment in the films. Our first analysis examined this question at the voxel level. In this analysis, the averaged time course of each PCA dimension (collapsed across all individuals in Sample 2) was included as a regressor of interest at the first level for each of the three movies for the brain activity recorded in each subject in Sample 1. To perform a group comparison of these analyses, we used FLAME in FSL with a cluster forming threshold of *z*=3.1 FWE, controlling for the number of regressors of interest to determine the significance of each cluster (p<0.0125). This generated four group-level thresholded maps, which we followed up with a FEAT query to extract cluster-wise parameter estimates, corresponding to regions whose activation during moments in the film was correlated with a specific thought pattern (see *Figure 4* and *Supplementary file 1f*). 'Episodic Knowledge' was significantly positively associated with activation in a region of dorsal visual cortex (*b*=0.62, 95% CI [0.27, 0.97]). 'Intrusive Distraction' was significantly associated with deactivation in the FPN (*b*=-0.78, 95% CI [-1.37,-0.20]). 'Verbal Detail' was significantly associated with suppression of activity in primary auditory cortex (*b*=-1.64, 95% CI [-2.11,-1.17]). Lastly, 'Sensory Engagement' was significantly associated with activation in both visual and auditory cortexes (*b*=1.26, 95% CI [0.81, 1.70]) (see *Supplementary file 1f* for the table of average Gradient score for each movie derived from this analysis). We also performed a functional connectivity analysis using each set of clusters as the seed region, see *Figure 4—figure supplement 1*, *Figure 4—figure supplement 2*, *Figure 4—figure supplement 3*, and *Supplementary file 1g*.

Our voxel-space analysis highlights two notable features of how thought patterns during movie-watching were linked to brain activity. First, most regions whose activity we can predict based on mDES scores tended to fall within sensory cortex. Notably, 'Sensory Engagement', a pattern of multi-sensory thought linked to sounds and images, is associated with increased activity in both the visual

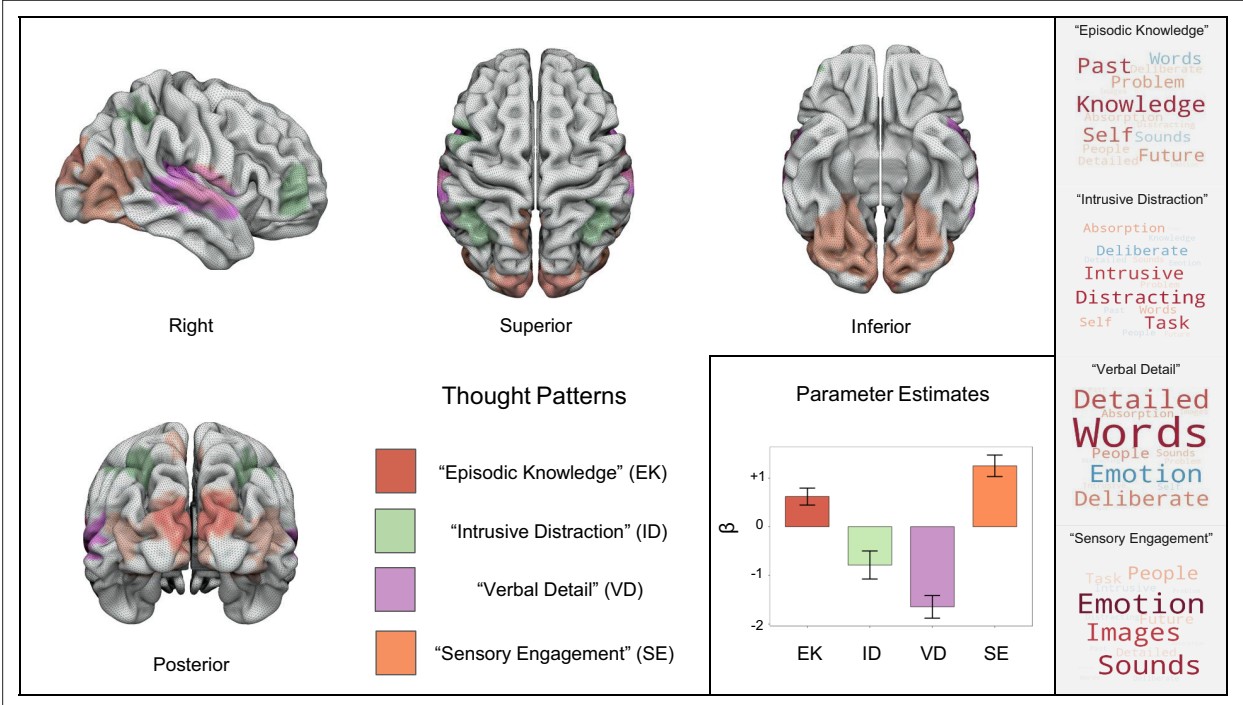

**Figure 4.** Group-level neural activation patterns associated with each of the dimensions of thought identified in a voxel space analysis. *Left to Right* - Regions in red are associated with activity corresponding to reports of 'Episodic Knowledge', green regions are associated with 'Intrusive Distraction', areas in purple are associated with 'Verbal Detail', and the regions in orange represent activity associated with 'Sensory Engagement'. The bar plot illustrates the directionality of each parameter estimate with error bars representing 95% Confidence Intervals. Corresponding word clouds for each thought pattern are presented on the right for reference (*Top to Bottom:* 'Episodic Knowledge', 'Intrusive Distraction', 'Verbal Detail', and 'Sensory Engagement').

The online version of this article includes the following figure supplement(s) for figure 4:

**Figure supplement 1.** Functional connectivity summary.

**Figure supplement 2.** Relationship of functional connectivity & Yeo 7 parcellation (DMN).

**Figure supplement 3.** Neurosynth decoding and functional connectivity relationship.

**Figure supplement 4.** Relationship of Intrusive Distraction map and FPN.

and auditory systems. Interestingly, these regions overlap with regions linked to 'Episodic Knowledge' maps (Posterior [orange and red]) and those linked to 'Verbal Detail' (Right [orange and purple]). Notably, since both Episodic Knowledge and Sensory Engagement show positive links to comprehension and greater activity in sensory cortex regions, these results support the hypothesis that perceptual coupling is an important feature of making sense of events during movie-watching (e.g. *Smallwood, 2013a*). Second, the only regions identified outside sensory cortex were linked to 'Intrusive Distraction' and broadly fall within regions of the FPN. *Figure 4—figure supplement 4* compares the regions linked to 'Intrusive Distraction' with the FPN as defined by *Yeo et al., 2011*, showing that the regions showing less activation during moments when Intrusive Distraction was high generally fall within this system. This pattern is consistent with views of the FPN as playing an active role in maintaining a state of non-distracted task focus (*Scolari et al., 2015*). See *Supplementary file 1h* for the analysis output and *Figure 5—figure supplement 1*, which shows each map is shown separately.

Our analysis highlighted significant overlap across analyses in visual and auditory cortex regions. To better understand the likely functions of these common regions, we calculated the overlap in these maps (left-hand panel of *Figure 5*) and performed a large-scale automated analysis consisting of over 4400 studies using Neurosynth, with the aim of identifying the most likely functions ascribed to these regions by prior research (See *Supplementary file 1i* for the specific loadings for each term). The results of this analysis are displayed in the form of word clouds where regions common to reduced 'Verbal Detail' and greater 'Sensory Engagement' are linked to auditory processes ('sounds', 'noise', and 'pitch'). In contrast, regions common to 'Sensory Engagement' and 'Episodic Knowledge' are

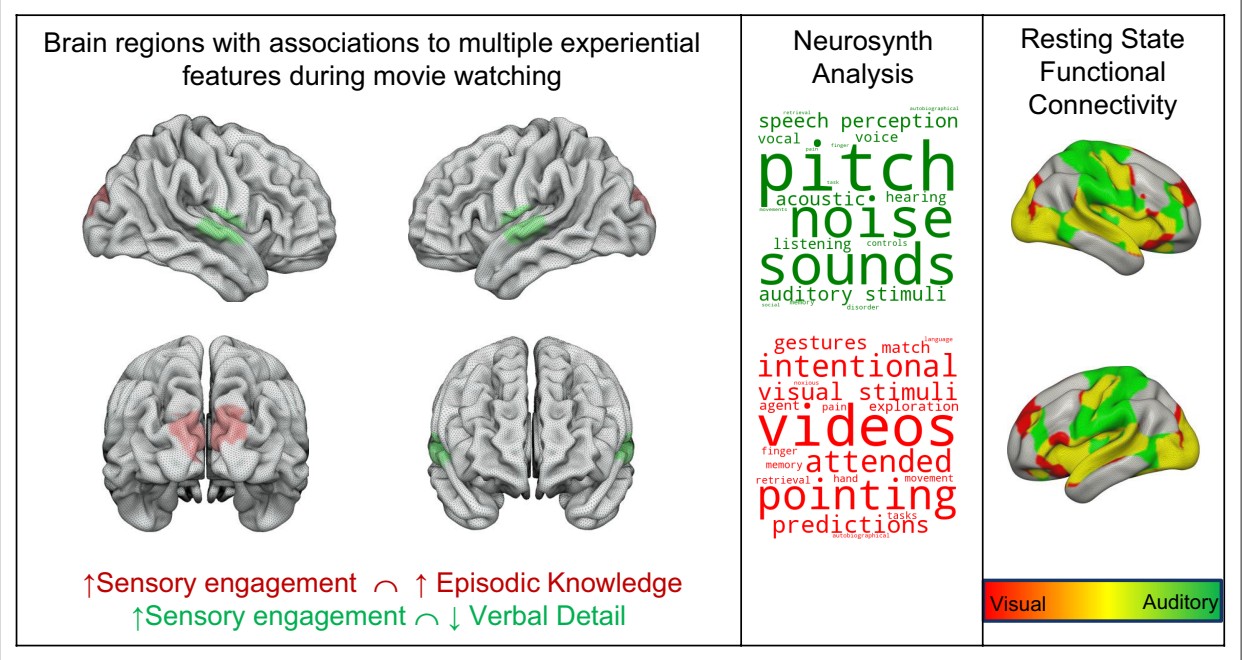

**Figure 5.** Brain regions associated with multiple experiential features during movie-watching. A region of superior temporal cortex is associated with positive reports of thoughts like 'Sensory Engagement' and negative reports of thoughts like 'Verbal Detail' (coloured green). A region of dorsal visual cortex was associated with both thoughts reported like 'Sensory Engagement' and 'Episodic Detail' (coloured red). The word clouds in the middle panel show the results of a Neurosynth analysis of the regions, highlighting the most likely functions associated with these regions. The font size describes their importance (bigger = more important), and the colour describes their polarity (darker = positive). The panel on the right shows the results of seed-based functional connectivity analysis of these regions of overlap from a separate resting-state study. Regions in red indicate those connected to the region of visual cortex, regions in green show those linked to auditory cortex, and regions in yellow are common to both spatial maps.

The online version of this article includes the following figure supplement(s) for figure 5:

**Figure supplement 1.** Summary of voxel-space brain maps by thought pattern.

most likely associated with 'videos', providing independent meta-analytic corroboration that these regions are paramount for movie-watching. The Neurosynth analysis, therefore, shows that the regions highlight by our analysis tend to be involved in sensory processing, and, are most commonly observed during movie-watching. Finally, we conducted a resting state functional connectivity analysis using the regions overlapping as seeds (right-hand panel of *Figure 5*). This highlighted that both regions exhibited functional connectivity patterns, including many overlapping areas (coloured yellow). Notably, both functional connectivity maps contained the seed regions of the other analysis.

## State space analysis

Our next analysis used a 'state-space' approach to determine how brain activity at each moment in the film predicted the patterns of thoughts reported at these moments (for prior examples in the domain of tasks, see *Mckeown et al., 2023*; *Turnbull et al., 2020*, See Methods). In this analysis, we used the coordinates of the group average of each TR in the 'brain space' and the coordinates of each experience sampling moment in the 'thought space'. To clarify, the location of a moment in a film in 'brain space' is calculated by projecting the grand mean of brain activity for each volume of each film against the first five dimensions of brain activity from a decomposition of the Human Connectome Project (HCP) resting state data, referred to as Gradients 1–5. 'Thought space' is the decomposition of mDES items to create thought pattern components, referred to as 'Episodic Knowledge', 'Intrusive Distraction', 'Verbal Detail', and 'Sensory Engagement'. We ran four LMMs, one for each thought component, in each case using the location of each sampling point in the movie on Gradients 1–5 as explanatory variables and the scores for each thought pattern component ('Episodic Knowledge', 'Intrusive Distraction', 'Verbal Detail' and 'Sensory Engagement') as dependent variables. Participant was included as a random intercept. The significance threshold was adjusted using the FDR to control for FWE within each model, controlling for five brain dimensions. After correction, we found two

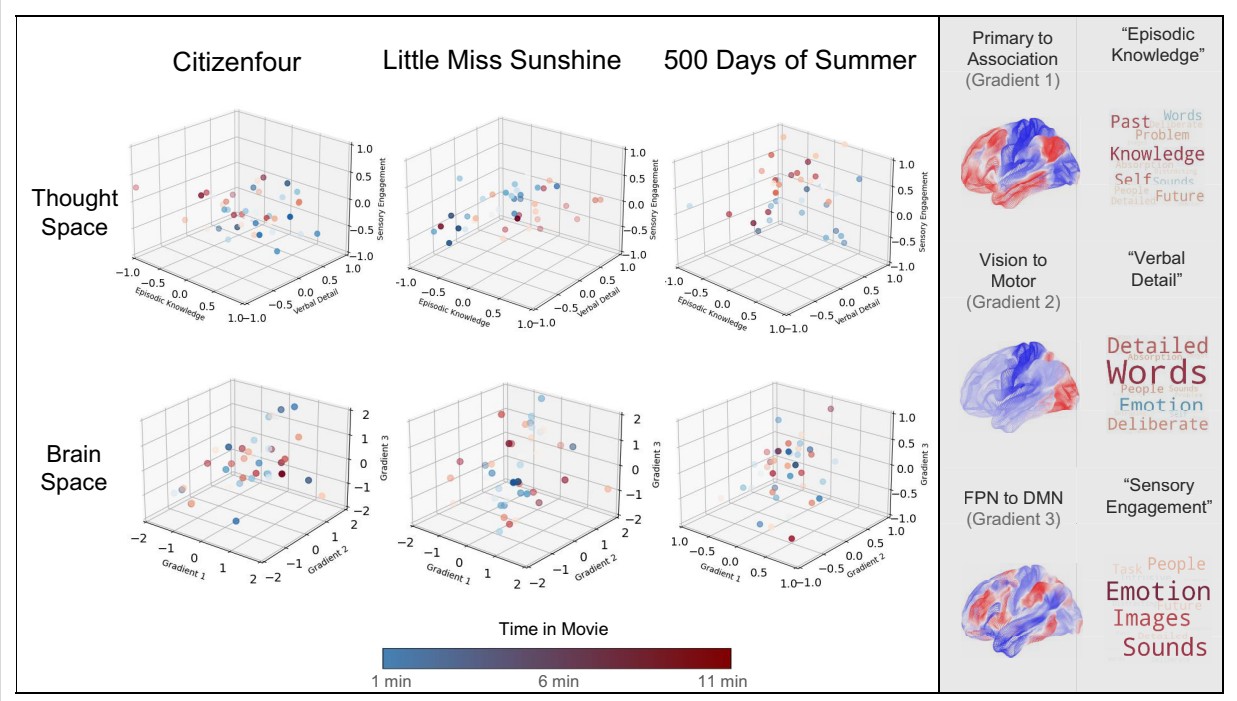

**Figure 6.** Comparison of the locations of each moment across the movie clip in the (top row) 'Thought Space' and the 'Brain Space'. *Left to Right* – 3D scatterplots of the coordinate locations of each thought pattern ('Episodic Knowledge', 'Verbal Detail', and 'Sensory Engagement') and gradients 1–3 (Gradient 1 Associated – Primary), Gradient 2 (Visual – Somato-motor), Gradient 3 (Frontoparietal – Default) during *Citizenfour*, *Little Miss Sunshine*, and *500 Days of Summer*. Observations in blue occur earlier during the film, and observations in red occur later in the film. The gradient maps (1-3) and thought pattern word clouds are presented on the right for reference.

significant main effects. First, we found a significant main effect of Gradient 4 (DAN to Visual), which predicted the similarity of answers to the 'Episodic Knowledge' component, $t(2046)=2.17$, $p=0.013$, $\eta 2=0.01$. This suggests that moments when thoughts were most similar to 'Episodic Knowledge' were associated with moments when activity was high in visual cortex and lower in regions of the dorsal attention network (See *Figure 6*). There was also a significant main effect of Gradient 1 (Primary to Association) predicting patterns of thought related to 'Sensory Engagement', $t(2046.34)=-3.26$, $p=0.006$, $\eta 2=0.01$. These results show that moments when thoughts are high on 'Sensory Engagement' were associated with increased brain activity in regions within the primary cortex low on Gradient 1 (see *Figure 6*). See *Supplementary file 1j* for complete results.

Our study highlighted links between patterns of self-reports resembling 'Sensory Engagement' and 'Episodic Knowledge' that were associated with brain activity patterns in our voxel and state space analyses. Therefore, in our final analysis, we aimed to understand the overlap between these two complementary approaches to understand the mapping between brain activity and experience during movie-watching. To this end, we used a spin test to formally understand the mapping between the voxel-based and state space analyses (*Alexander-Bloch et al., 2018*). To this end, we sampled the location of the identified cluster in our voxel analysis on the gradient of interest (e.g. the cluster of voxels associated with 'Sensory Engagement' on Gradient 1) and used spin tests (*Alexander-Bloch et al., 2018*) to determine the likelihood that a score with this magnitude would occur by chance (based on a null distribution of 2500 permutations). This analysis identified that our voxel-based estimate of 'Sensory Engagement' falls within regions of sensory cortex implied by the state space analysis (i.e. the sensory end of Gradient 1) at a level that is unlikely to occur by chance, $p=0.018$ (two-tailed). In contrast, the location of 'Episodic Knowledge' on Gradient 4 was not significant, $p=0.251$ (see *Figure 7*). This analysis indicates that for the 'Sensory Engagement' component both the voxel-space and state-space analysis yielded comparable results highlighting common regions of sensory and auditory cortex.

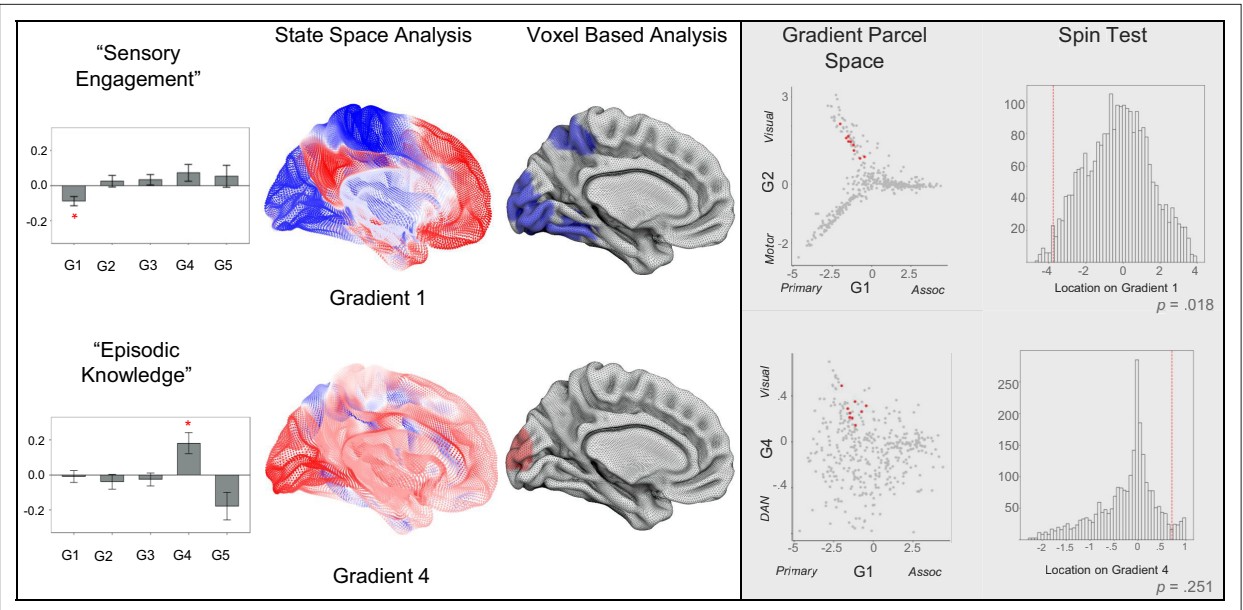

**Figure 7.** Comparison of 'State-Space' and Voxel based analyses of 'Sensory Engagement' and 'Episodic Knowledge' with Gradients. *Left to right* – The barplots illustrate the associations for the significant models using Gradients 1–5 as explanatory variables and the thought patterns, 'Sensory Engagement' and 'Episodic Knowledge' as dependent variables. We performed two spin tests to formally compare these results to those using the voxel space analysis (permutation = 2500). The spin tests revealed the location of the cluster of voxels associated with 'Sensory Engagement' are located within the sensory regions of Gradient 1, unlikely to have occurred by chance, p=0.018. In contrast, the location of the cluster of voxels associated with Episodic Knowledge on Gradient 4 was within the null distribution, p=0.251. The locations of the relevant clusters in gradient parcel space are presented in the scatter plots (red points indicate the location of parcels from the relevant comparison).

## Discussion

Our study aimed to identify how patterns of thought during movie-watching relate to brain activity during movie clips from three different films: *Citizenfour* (a documentary), *Little Miss Sunshine* (a comedy), and *500 Days of Summer* (a romance). We used open-source fMRI data from one group of participants (Sample 1) who watched these films while brain activity was recorded using fMRI. We then measured ongoing thought patterns using mDES in a second group of participants (Sample 2) for whom no brain activity was acquired (*Figure 1*). We used a novel sampling approach to build a detailed description of the time series of different thought patterns every 15 s in the clips while only sampling individual participants a relatively small number of times per movie, minimizing disruption of the subjective experience of movie-watching. Our analyses examined the overlap between the time series of brain activity in one group of participants (Sample 1) and reported thought patterns in a second group of participants (Sample 2) to reveal the relationship between brain activity at different moments in a film and the associated experiential states.

Across the movies, we identified four thought patterns. First, 'Episodic knowledge' was linked to experiences related to knowledge, the past, and the self. This pattern was also highest during the romance movie, specifically associated with better memory of information in this context and increased activity in dorsal medial regions of visual cortex by our state space and voxel space analysis. Second, 'Intrusive Distraction' was related to thoughts with intrusive, distracting features that were spontaneous in nature. This thought pattern predicted poorer overall comprehension across all the movies, was higher in the documentary, and emerged in moments during movies associated with reduced activation in regions of the FPN by our voxel space analysis. Third, 'Verbal Detail', which described experience as deliberate, detailed experiences in the form of words and with a negative emotional valence, was most prevalent in the documentary and associated with relative reductions in auditory cortex activation using our voxel space analysis. Finally, 'Sensory Engagement' was related to multi-modal sensory experience (loading on images, sounds, and people with a positive emotional tone). 'Sensory Engagement' was highest in the romance movie, associated with better

comprehension performance across all movies, linked to activity in sensory cortex by both the voxel-based analysis and the state-space analysis, which were formally linked through a spin test.

Our study supports the hypothesis that perceptual coupling between the brain and external input is a core feature of making sense of events in movies (e.g. *Smallwood, 2013b*). For example, 'Sensory Engagement', a pattern of enjoyable multi-sensory experience, was linked to better memory for information across all the movies and emerged when activity was high in both auditory and visual cortexes (regions at the sensory end of the principle gradient of functional brain organization, *Margulies et al., 2016*). 'Sensory Engagement' was the thought pattern with the most consistent and most apparent links to the brain since it was the only thought pattern that showed a brain-thought mapping across our voxel- and state-space analysis that were formally linked using a spin test (see *Figures 2 and 6*). Similarly, reports of 'Episodic Knowledge' emerged when brain activity was high within a dorsal region of visual cortex and was linked to better comprehension in one of the films (*500 Days of Summer*). Together, these data provide important corroboration for the hypothesis that states of sensory coupling support better memory for environmental events (*Smallwood et al., 2021b*; *Smallwood, 2013a*). Further, they also provide support for contemporary perspectives that movie-watching is a useful and important paradigm for understanding the brain basis behind naturalistic states because it allows brain function to be understood through the lens of a state rich in complex sensory input (*Finn, 2021*).

Our study also provides support for the hypothesized role the frontoparietal system plays in supporting states of non-distracted focus during movie-watching. Reports of 'Intrusive Distraction' were the only thought pattern associated with activity outside primary sensory systems and was seen to emerge at moments in films when activity during regions within the FPN was reduced See *Figure 4—figure supplement 4* for the overlap between the regions identified linked to 'Intrusive Distraction' and the FPN as defined by Yeo and colleagues (*Yarkoni et al., 2011*). Interestingly, the association between greater distraction and reduced activity within the FPN is consistent with this network's assumed role in goal maintenance (*Cole and Schneider, 2007*). This result also confirms predictions from psychological research that states of distraction, like mind-wandering, often emerge when executive control is reduced (*McVay and Kane, 2010*). This hypothesis gains further support for the consistent negative association with comprehension. Notably, 'Intrusive Distraction' was the only component that showed no evidence of temporal variation across the movie clips we sampled. It is possible, therefore, that processes that drive the occurrence of states such as 'Intrusive Distraction' are likely to depend on individual and contextual factors (e.g. poor executive control *McVay and Kane, 2010*; *McVay and Kane, 2012*) and/or intrinsic changes in brain activity. Recently, a study using intra-cranial recordings established that periods of distraction, with similar features to those observed in this study, occur when sharp wave ripples within the hippocampus are common (*Iwata et al., 2024*). It is likely, therefore, that further research may be necessary to understand the brain-cognition mappings which lead to distraction. For example, in the future, we can systematically explore conditions in which this state is more or less present and highlight deviations from the mean as markers to better understand states of cognition that are less related to a state of perceptual coupling than are other features of movie-watching. Notably, using mDES, we can also identify the same thought pattern in our analysis in daily life research (*Mckeown et al., 2021*; *Mulholland et al., 2023*), suggesting this thought pattern occurs in situations beyond movie-watching. Further investigation using mDES, for example with other forms of media, and other methods of brain imaging, can improve our understanding of what lead to the onset of distracted states.

Although our study highlighted neural activity in sensory cortex and regions of association cortex with the frontoparietal system, we found less evidence for the hypothesized role of the DMN during the movie-watching experience. Notably, the pattern of 'Episodic Knowledge' identified by our analysis focuses on features of cognition such as knowledge, people, and oneself — all of which are terms that previous literature suggests could relate to the DMN (*Smallwood et al., 2021b*; *Yang et al., 2023*). However, despite this conceptual mapping, neither our voxel space nor our state space analysis highlighted that this experience was related to moments when brain activity was higher within the DMN (See *Figures 3 and 6*).

There are several possible methodological reasons why such a mapping within the DMN may nonetheless exist. For example, our choice of films (documentary, romance, and comedy) may have precluded a genre in which the DMN may play a more obvious role (e.g. mystery or suspense). Another

possibility is that the DMN may be relevant to an understanding aspect of experience that is only captured during longer intervals of movie-watching, such as extended plot lines, unexpected events, or other features of movies that depend on the segmentation of a movie into different events (*Geerligs et al., 2022*). We only sampled experience in short 10 min clips, so the DMN may relate to aspects of experience that are important for movie-watching over longer time periods. It is also possible that the unique features of the DMN make it difficult for our method to reveal its role in experience. The DMN is a spatially heterogeneous system and highly variable across individuals (*Braga and Buckner, 2017*). Since our analytic approach links thought patterns in one set of individuals to brain activity in another, it could be challenging for this method to identify its role in movie-related thought patterns in a highly idiosyncratic brain network such as the DMN (*Braga and Buckner, 2017*; *Daitch and Parvizi, 2018*). This possibility could be easily tested by examining mappings between thought patterns and individuals using precision scanning methods (*Gordon et al., 2017*). Studies have also highlighted that the DMN is heterogeneous in the functions it is involved in and, in particular, is hypothesized to shift flexibly from perpetually decoupled to coupled states (*Zhang et al., 2022*). So, for example, the role this network is hypothesized to play in off-task or mind-wandering states (e.g. *Christoff et al., 2009*) may obscure its' role in perceptually coupled states (like movie-watching).

It is important to note that while our study does not establish what role DMN plays in movie-watching states, it does highlight a clear role for sensory systems in experiential states that are time-locked, or 'coupled', to events in the film. Thus, based on our study, whatever role the DMN plays during movie-watching, it is likely to be built upon the foundational role sensory systems play in our thoughts and feelings while we watch films. Consistent with this possibility, contemporary views on the DMN argue that its function arises from its' topographical location in the cortex (*Smallwood et al., 2021b*). According to this perspective, the DMN is located at the maximal distance from primary systems but also constitutes the apex of processing streams (like the ventral and dorsal streams *Margulies et al., 2016*). We have previously argued that whatever role the DMN plays in cognition may entail interactions with primary systems, possibly through the transformation of neural signals along different processing streams (*Smallwood et al., 2021b*). In other words, it is possible that the DMN plays a role in movie-watching that complements information processing in sensory input, an important but possibly less direct contribution to the movie-watching experience than regions in the visual or auditory cortex. Since the DMN is widely hypothesized to be important in movie-watching, we performed an exploratory functional connectivity analysis to examine whether the sensory regions we identified in our study are functionally coupled to the DMN at rest (see *Figure 4—figure supplement 2*). This revealed that sensory regions identified in our study shared a common set of regions within the DMN (including anterior regions of the temporal lobe and the inferior frontal gyrus; see *Figure 4—figure supplement 2* and *Figure 4—figure supplement 3*). This analysis was exploratory, so any results should be treated with caution; however, it is consistent with the possibility that a more fine-grained precision mapping approach could identify the role these regions play in ongoing thought during movie-watching (*Gordon et al., 2017*).

In summary, our study used a novel paradigm to establish the role primary systems play during our experiences while we watch movies. Nonetheless, important questions about other features of experience during move-watching remain unanswered. For example, patterns of 'Verbal Detail' were associated with moments in the films where auditory cortex activation was reduced. This may reflect a shift in attention away from the processing of the auditory input related to the movie towards evaluative thoughts about the people or events in the film, perhaps in the form of inner speech (*Mckeown et al., 2023*). These thoughts may occur when participants form opinions about movie characters, elaborate on the context, or make inferences about the information they have encoded from the film (*Alderson-Day and Fernyhough, 2015*). This possibility could be easily explored by examining more specific experience sampling items that directly target inner speech or comprehension questions that target inferential processing on events within the movies (See *Smallwood et al., 2008b*).

Importantly, our study provides a novel method for answering these questions and others regarding the brain basis of experiences during films that can be applied simply and cost-effectively. As we have shown, mDES can be combined with existing brain activity, allowing information about both brain activity and experience to be determined at a relatively low cost. For example, the cost-effective nature of our paradigm makes it an ideal way to explore the relationship between cognition and neural activity during movie-watching during different genres of film. In neuroimaging, conclusions are often

made using one film in naturalistic paradigm studies (*Yang et al., 2023*). Although the current study only used three movie clips, restraining our ability to form strong conclusions regarding how different patterns of thought relate to specific genres of film, in the future, it will be possible to map cognition across a more extensive set of movies and discern whether there are specific types of experience that different genres of films engage. One of the major strengths of our approach, therefore, is the ability to map thoughts across groups of participants across a wide range of movies at a relatively low cost.

Nonetheless, this paradigm is not without limitations. This is the first study, as far as we know, that attempts to compare experiential reports in one sample of participants with brain activity in a second set of participants, and while the utility of this method enables us to understand the relationship between thought and brain activity during movies, it will be important to extend our analysis to mDES data during movie-watching while brain activity is recorded. In addition, our study is correlational in nature, and in the future, it could be useful to generate a more mechanistic understanding of how brain activity maps onto the participants experience. Our analysis shows that mDES is able to discriminate between films, highlighting its broad sensitivity to variation in semantic or affective content. Armed with this knowledge, we propose that in the future, researchers could derive mechanistic insights into how the semantic features may influence the mDES data. For example, it may be possible to ask participants to watch movies in a scrambled order to understand how the structure of semantic or information influences the mapping between brains and ongoing experience as measured by mDES. Finally, our study focused on mapping group-level patterns of experience onto group-level descriptions of brain activity. In the future it may be possible to adopt a 'precision-mapping' approach by measuring longer periods of experience using mDES and determining how the neural correlates of experience vary across individuals who watched the same movies while brain activity was collected (*Gordon et al., 2017*). In the future, we anticipate that the ease with which our method can be applied to different groups of individuals and different types of media will make it possible to build a more comprehensive and culturally inclusive understanding of the links between brain activity and movie-watching experience.

Finally, it is worth considering whether the patterns of brain activity identified by our analysis reflect the stimuli that are processed during movie watching, or the cognitive and affective processing of this information. On the one hand, the regions we found were often within regions of sensory cortex, areas of the brain which are often ascribed basic stimulus processing functions (*Kaas and Collins, 2001*). Moreover, according to perspectives on cognition derived from more traditional task paradigms, complex features of cognition, such as the regulation of thought, are often attributed to regions of association cortex, such as the dorsolateral prefrontal cortex (*Turnbull et al., 2019a*). It is possible, based on these views that the identification of regions of visual and auditory cortex by our study reflects the participants attention to sensory input, rather than the complex analysis of these inputs that may be required for certain features of the movie watching experience. On the other hand, it is possible that the movie-watching state is a qualitatively different type of mental state to those that emerge in typical task situations. For example, unlike tasks, the movie-watching state is characterized by multi-modal sensory input, semantically rich themes, that evolve together to reveals a continuous narrative to the viewer. It is possible, therefore, that these features allow movies to engender a situation an absorbed state where a relatively higher amount of processing are achieved in sensory cortex than would occur in traditional task paradigms (when systems in association cortex may be needed to maintain information related to task rules). Important headway into addressing this uncertainty can be achieved by using mDES to compare the types of states that occur in different contexts (including both movies and tasks) and comparing the topography of brain activity associated with different states.

## Methods
### Participant pool – laboratory sample
The sample consisted of 120 participants (98 women (81.7%), 17 men (14.2%), 5 non-binary or similar gender identity (3.3%); age: *M*=18.83, SD = 1.19, range of 18–23) who participated in the in-person laboratory study to watch three 11 min movie clips, responded to mDES probes, and completed a brief comprehension assessment. All participants spoke English, with 95% of the sample primarily residing in Canada (China [1.7%], India [0.8%], Nigeria [0.8%], USA [1.7%]). This study was granted

ethics clearance by the Queen's University General Research Ethics Board (#6036804). Participants were recruited between March 2023 and April 2023 through the Queen's University Psychology Participant Pool. Participants provided written, informed consent via electronic documentation before participating in the research study. Participants were rewarded with one-course credit or $10.00 for their participation and were provided with a verbal and written debrief form upon completion of the study. All data and corresponding analysis scripts included in this manuscript are available via Mendeley Data (DOI: 10.17632/mgb7ftwr9d.1).

## Participant pool – brain data sample

See Aliko and colleagues for a description of the sample (*Aliko et al., 2020*).

## Participant pool – resting-state sample

191 student volunteers (mean age = 20.1 ± 2.25 years, range 18–31; 123 females) with normal or corrected-to-normal vision and no history of neurological disorders participated in this study. Written informed consent was obtained from all subjects prior to the resting-state scan. The study was approved by the ethics committees of the Department of Psychology and York Neuroimaging Centre, University of York. Previous studies have used this data to examine the neural basis of memory and mind-wandering, including region-of-interest-based connectivity analysis and cortical thickness investigations (*Karapanagiotidis et al., 2017*; *Wang et al., 2018*; *Evans et al., 2020*; *Gonzalez Alam et al., 2018*; *Gonzalez Alam et al., 2019*; *Gonzalez Alam et al., 2022*; *Gonzalez Alam et al., 2021*; *Poerio et al., 2017*; *Turnbull et al., 2019b*; *Vatansever et al., 2017*).

## Procedure

Participants attended an individual in-person testing session at the laboratory at Queen's University to watch movie clips after providing written informed consent and basic demographic information. Participants were assigned to a testing booth, a small room with a desk, a chair, a computer to present the stimuli, and headphones to listen to the audio stimuli. Participants had to attend to the computer screen to watch and listen to three randomly presented 11 min video clips. During each movie clip, participants were briefly interrupted five times to answer randomly assigned mDES probes about the content of their thoughts just prior to the probe. After the first minute of each clip, each probe was delivered once every two minutes, using a jittered technique, by assigning participants to a counterbalanced probe order to minimize the systematic impact of prior and later probes at any given sampling moment (see *Figure 3—figure supplement 1* for visualization). Once participants finished watching the three clips, they completed a 12-item comprehension questionnaire on Qualtrics, with four items related to information from each of the three movie clips (see *Supplementary file 1d*).

## Multi-dimensional experience sampling (mDES)

Participants received 16 total mDES probes across the three clips, five for each, and all responses were made with respect to their thoughts just before the probe interrupted their viewing. No probes were administered within the first minute of the clip — the first possible probe was administered at the 75 s mark to allow participants to situate themselves with the context of the movie clip. Each of the 16 mDES questions appeared in a randomized order, and participants were asked to use the directional arrow keys to move a slider across the screen to indicate, on a scale of 1 (not at all) to 10 (completely), how much that particular feature characterized their thoughts. The specific items used in this experiment are presented in *Supplementary file 1a*.

## Probe orders

There were 16 probe orders that a participant could be assigned to for each movie, which determined the delivery time of the five mDES probes throughout each 11 min clip. Each subject ID was assigned to three different probe orders for each of the three films, and no subject ID was given the same probe order twice. This was achieved by creating a matrix of equally distributed probe orders across subject IDs to ensure each moment in the movie was probed an equal amount of times while uniquely distributing probes to control for order effects. Probe orders were designed to sample participants at every 15 s interval of the entire movie clip but only probed a single participant five times per clip. This allowed us to sample experience as frequently as possible without interrupting participants from

naturalistic viewing by oversampling or too frequent probes and to control for ordering effects from the delivery time of the other probes. Each participant received a probe approximately every two minutes using a jittered technique. The first eight probe orders do not share any of the same probe delivery times, whereas the latter eight probe orders (9-16) have been shuffled so that they share one probe time with only one of the orders from the former eight orders. Across orders 9–16, each probe from the first eight orders is repeated in a different combination of probes so that mDES responses at each probe are derived from participants in two different orders. See (*Figure 3—figure supplement 1*).

### Movie clip stimuli

Movie stimuli were presented in 11 min scenes from *Citizenfour*, *Little Miss Sunshine*, and *500 Days of Summer*. Stimuli were selected from the Naturalistic Neuroimaging Database (NNDb; *Aliko et al., 2020*) and chosen based on genre, and they were cut from the full-length movie down to 11 min clips. Participants were informed they would watch three movie clips from different genres (romance, comedy, documentary) but were presented randomly. Written instructions were presented on screen at the beginning of each clip. After watching the three clips and responding to the mDES probes, participants were presented with a Qualtrics questionnaire to complete a comprehension test on the content of each film clip.

### Comprehension questions

Participants completed a comprehension test of 12 questions, four from each movie. The questions were created collaboratively to test general knowledge about the movie that would otherwise not be common sense and cover events during the clip's beginning, middle, and end. An example of one of the comprehension questions was 'What breakfast item did Olive order a la mode?' for the movie clip *Little Miss Sunshine*. A table of all the questions with corresponding answers can be found in *Supplementary file 1d*. Participants responded using 1–2 words and were otherwise instructed to enter '?' if they had no answer.

### Brain analysis

Our analyses used brain data acquired and shared by the NNDb, an open-access database of pre-processed MNI 2 mm fMRI data (TR = 1) of participants who watched one of 10 full-length movies (*Aliko et al., 2020*). We utilized MNI 2 mm fMRI data corresponding to participants who watched *Little Miss Sunshine* (n=6), *Citizenfour* (n=18), and *500 Days of Summer* (n=20). The specific pre-processing steps applied to the brain data and specific details of the sample are described in *Aliko et al., 2020*.

### Voxel-space analysis

Our analysis used the pre-processed data from *Aliko et al., 2020*. The first step in our analysis was to extract the brain activity of each individual for the 10 min section that we sampled in experience using mDES. To map the mDES time series onto these data, we created a mean time series for each movie, which described the mDES experience, averaged across 40 observations at every 15 s interval. Next, we interpolated this time series to generate a time series of experiences that matched the TR used to sample brain activity in Sample 1 (1 s). Next, the interpolated time series for each PCA for each film were included as regressors for each individual's brain activity (i.e. four regressors for each movie). Finally, we used FLAME as implemented in FSL to perform a group-level analysis across the three movie clips. In this analysis, we set the cluster-forming threshold at z=3.1 and corrected for FWE by accounting for the number of voxels in the brain, the three movies we examined, and the four PCAs in each movie. This resulted in the correction of the FWE p-value from FSL p<0.0025.

### State-space analysis

To create the 'state-space' coordinates, we first calculated a group-averaged timeseries for each movie. To do this, we first z-scored each individual's timeseries data and calculated the mean activity in each voxel at each TR across the whole sample, resulting in a group-averaged brain volume at each TR. Next, we applied a binarized mask to each group-averaged brain volume at each TR. This mask was generated based on the (cortical and subcortical) gradient maps openly available on Neurovault (https://identifiers.org/neurovault.collection:1598). These gradient maps were produced from the

decomposition of the Human Connectome Project resting-state fMRI data (*Margulies et al., 2016*). Then, we calculated the (spearman rank) correlation between each group-averaged per-TR brain map and each of the first five gradient maps. Consistent with published literature, the results of these correlations constitute the coordinates of each moment of the film in the 5D Brain space (*Mckeown et al., 2023*). An example of these coordinates is presented in the upper left panel of *Figure 5*. The code for this analysis is openly available at https://github.com/willstrawson/StateSpace (v1.0.0), copy archived at *Mckeown et al., 2024* (https://zenodo.org/records/14112469).

## Cognitive decoding

Connectivity maps were uploaded to Neurovault (*Gorgolewski et al., 2015*; https://neurovault.org/collections/13821/) and decoded using Neurosynth (*Yarkoni et al., 2011*). Neurosynth is an automated analysis tool that uses text-mining approaches to extract terms from neuroimaging articles that typically co-occur with specific peak coordinates of activation. It can be used to generate a set of terms frequently associated with a spatial map. The results of cognitive decoding were rendered as word clouds using in-house scripts implemented in Python. We excluded terms referring to neuroanatomy (e.g. 'inferior' or 'sulcus'), as well as the second occurrence of repeated terms (e.g. 'semantic' and 'semantics'). The size of each word in the word cloud relates to the frequency of that term across studies.

## Analysis of intrinsic functional connectivity using resting-state fMRI

Our analysis additionally used resting state cohort data (see below) to seed the maps created from each of the four thought pattern time series regressors from the prior analysis. We used this seed-based analysis to see if different resting state networks converge with the maps we have generated from movie-watching.

## Pre-processing

Pre-processing and statistical analyses of resting-state data were performed using the CONN functional connectivity toolbox V.20a (http://www.nitrc.org/projects/conn; *Whitfield-Gabrieli and Nieto-Castanon, 2012*) implemented through SPM (Version 12.0) and MATLAB (Version 19 a). For pre-processing, functional volumes were slice-time (bottom-up, interleaved) and motion-corrected, skull-stripped and co-registered to the high-resolution structural image, spatially normalized to the Montreal Neurological Institute (MNI) space using the unified-segmentation algorithm, smoothed with a 6 mm FWHM Gaussian kernel, and band-passed filtered (0.008–0.09 Hz) to reduce low-frequency drift and noise effects. A pre-processing pipeline of nuisance regression included motion (12 parameters: the six translation and rotation parameters and their temporal derivatives), scrubbing (outlier volumes were identified through the composite artifact detection algorithm ART in CONN with conservative settings, including scan-by-scan change in global signal z-value threshold = 3; subject motion threshold = 5 mm; differential motion and composite motion exceeding 95% percentile in the normative sample) and CompCor components (the first five) attributable to the signal from white matter and CSF (*Behzadi et al., 2007*) as well as a linear detrending term, eliminating the need for global signal normalization (*Chai et al., 2012*; *Murphy et al., 2009*).

## Seed selection and analysis

Intrinsic connectivity seeds were binarized masks derived from voxel-space analysis using FLAME through FSL. We excluded all non-grey matter voxels that fell within these masks. We performed seed-to-voxel analyses convolved with a canonical hemodynamic response function for each seed. At the group-level, analyses were conducted using CONN with cluster correction at p<0.05 and a threshold of p-FDR=0.001 (two-tailed) to define contiguous clusters.

## Additional information

### Funding

| Funder | Grant reference number | Author |
| --- | --- | --- |
| Natural Sciences and Engineering Research Council of Canada | #RGPIN 2023-03496 | Jonathan Smallwood |

The funders had no role in study design, data collection and interpretation, or the decision to submit the work for publication.

### Author contributions

Raven Star Wallace, Conceptualization, Resources, Data curation, Formal analysis, Validation, Investigation, Visualization, Methodology, Writing – original draft, Project administration, Writing – review and editing; Bronte Mckeown, Philippe Forest, Formal analysis, Writing – review and editing; Ian Goodall-Halliwell, Resources, Methodology, Writing – review and editing; Louis Chitiz, Validation, Writing – review and editing; Theodoros Karapanagiotidis, Formal analysis, Validation, Writing – original draft, Writing – review and editing; Bridget Mulholland, Samyogita Hardikar, Hao-Ting Wang, Will Strawson, Michael Milham, Ting Xu, Writing – review and editing; Adam Turnbull, Tamara Vanderwal, Boris C Bernhardt, Daniel S Margulies, Giulia L Poerio, Elizabeth Jefferies, Jeffrey D Wammes, Conceptualization, Writing – review and editing; Tirso RJ Gonzalez Alam, Data curation, Formal analysis, Writing – original draft, Writing – review and editing; Jeremy I Skipper, Conceptualization, Resources, Writing – review and editing; Robert Leech, Conceptualization, Software, Formal analysis, Visualization, Writing – review and editing; Jonathan Smallwood, Conceptualization, Resources, Supervision, Funding acquisition, Validation, Investigation, Visualization, Writing – original draft, Project administration, Writing – review and editing

### Author ORCIDs

Raven Star Wallace ⬛ https://orcid.org/0009-0003-0414-0254
Samyogita Hardikar ⬛ https://orcid.org/0000-0003-4380-5055
Tirso RJ Gonzalez Alam ⬛ https://orcid.org/0000-0003-4510-2441
Boris C Bernhardt ⬛ https://orcid.org/0000-0001-9256-6041
Jeremy I Skipper ⬛ https://orcid.org/0000-0002-5503-764X
Jeffrey D Wammes ⬛ https://orcid.org/0000-0002-8923-5441
Robert Leech ⬛ https://orcid.org/0000-0002-5801-6318
Jonathan Smallwood ⬛ http://orcid.org/0000-0002-7298-2459

### Ethics

Human subjects: Active, informed consent to participate, obtain, and publish anonymized data was collected from each participant. The specific ethical approval we obtained were derived from the guidelines set out by Queen's University General Research Ethics Board (GREB) (#6036804).

Reviewer #1 (Public review): https://doi.org/10.7554/eLife.97731.4.sa1
Reviewer #2 (Public review): https://doi.org/10.7554/eLife.97731.4.sa2
Reviewer #3 (Public review): https://doi.org/10.7554/eLife.97731.4.sa3
Author response https://doi.org/10.7554/eLife.97731.4.sa4

## Additional files

### Supplementary files

Supplementary file 1. Supplementary table data for corresponding manuscript results. **(a)** Multidimensional Experience sampling (mDES). **(b)** Percent variance explained by principal components by movie **(c)** Linear Mixed Models of Variance in Thoughts across Movies. **(d)** Movie Comprehension Questions. **(e)** Linear Mixed Models of Comprehension model. **(f)** FSL FEAT Query Parameter Estimates. **(g)** Grand average of Gradient score by movie. **(h)** Functional Connectivity Cluster Analysis (FLAME). **(i)** Neurosynth Decoder Analysis. **(j)** Linear Mixed Models of Gradients 1–5 Fixed Effects for each Thought Pattern

MDAR checklist

## Data availability

All data and corresponding analysis scripts included in this manuscript are available via Mendeley Data (https://doi.org/10.17632/mgb7ftwr9d.1).

The following dataset was generated:

| Author(s) | Year | Dataset title | Dataset URL | Database and Identifier |
|---|---|---|---|---|
| Wallace RS | 2024 | Mapping patterns of thought onto brain activity during movie-watching Data and Scripts | https://doi.org/10.17632/mgb7ftwr9d.1 | Mendeley Data, 10.17632/mgb7ftwr9d.1 |

The following previously published datasets were used:

| Author(s) | Year | Dataset title | Dataset URL | Database and Identifier |
|---|---|---|---|---|
| Aliko S, Huang J, Gheorghiu F, Meliss S, Skipper JI | 2021 | Naturalistic Neuroimaging Database | https://doi.org/10.18112/openneuro.ds002837.v2.0.0 | OpenNeuro, 10.18112/openneuro.ds002837.v2.0.0 |
| Karapanagiotidis T, Bernhardt BC, Jefferies E, Smallwood J | 2016 | Tracking thoughts: Exploring the neural architecture of mental time travel during mind-wandering | https://identifiers.org/neurovault.collection:1448 | NeuroVault, 1448 |

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
