## [Editor Report · eLife Assessment]

This study presents a **valuable** methodological advancement in quantifying thoughts over time. A novel multi-dimensional experience-sampling approach is presented, identifying data-driven patterns that the authors use to interrogate fMRI data collected during naturalistic movie-watching. The experimentation is inventive and the analyses carried out and results presented are **convincing**.

---

## [Referee Report · Reviewer #1 (Public review)]

The authors used a novel multi-dimensional experience sampling (mDES) approach to identify data-driven patterns of experience samples that they use to interrogate fMRI data collected during naturalistic movie-watching data. They identify a set of multi-sensory features of a set of movies that delineate low-dimensional gradients of BOLD fMRI signal patterns that have previously been linked to fundamental axes of cortical organization.

---

## [Referee Report · Reviewer #2 (Public review)]

The present study explores how thoughts map onto brain activity, a notoriously challenging question because of the dynamic, subjective, and abstract nature of thoughts. To tackle this question, the authors collected continuous thought ratings from participants watching a movie, and additionally made use of an open-source fMRI dataset recorded during movie watching as well as five established gradients of brain variation as identified in resting state data. Using a voxel-space approach, the results show that episodic knowledge, verbal detail, and sensory engagement of thoughts commonly modulate visual and auditory cortex, while intrusive distraction modulates the frontoparietal network. Additionally, sensory engagement mapped onto a gradient from primary to association cortex, while episodic knowledge mapped onto a gradient from the dorsal attention network to visual cortex. Building on the association between behavioral performance and neural activation, the authors conclude that sensory coupling to external input and frontoparietal executive control are key to comprehension in naturalistic settings.

The manuscript stands out for its methodological advancements in quantifying thoughts over time and its aim to study the implementation of thoughts in the brain during naturalistic movie watching.

Strengths:

(1) The study raises a question that has been difficult to study in naturalistic settings so far but is key to understanding human cognition, namely how thoughts map onto brain activation.

(2) The thought ratings introduce a novel method for continuously tracking thoughts, promising utility beyond this study.

(3) The authors used diverse data types, metrics, and analyses to substantiate the effects of thinking from multiple perspectives.

---

## [Referee Report · Reviewer #3 (Public review)]

This study attempted to investigate the relations between processing in the human brain during movie watching and corresponding thought processes. This is a highly interesting question, as movie watching presents a semi-constrained task, combining naturally occurring thoughts and common processing of sensory inputs across participants. This task is inherently difficult because in order to know what participants are thinking at any given moment, one has to interrupt the same thought process which is the object of study.

This study attempts to deal with this issue by aggregating staggered experience sampling data across participants in one behavioral study and using the population level thought patterns to model brain activity in different participants in an open access fMRI dataset.

The behavioral data consist of 120 participants who watched 3 11-minute movie clips. Participants responded to the mDES questionnaire: 16 visual scales characterizing ongoing thought 5 times, two minutes apart, in each clip. The 16 items are first reduced to 4 factors using PCA, and their levels are compared across the different movies. The factors are "episodic knowledge", "intrusive distraction", "verbal detail", and "sensory engagement". The factors differ between the clips, and distraction is negatively correlated with movie comprehension and sensory engagement is positively correlated with comprehension.

The components are aggregated across participants (transforming single subject mDES answers into PCA space and concatenating responses of different participants) and are used as regressors in a GLM analysis. This analysis identifies brain regions corresponding to the components. The resulting brain maps reveal activations that are consistent with the proposed mental processes (e.g. negative loading for intrusion in frontoparietal network, positive loadings for visual and auditory cortices for sensory engagement).

Then, the coordinates for brain regions which were significant for more than one component are entered into a paper search in neurosynth. It is not clear what this analysis demonstrates beyond the fact that sensory engagement contained both visual and auditory components.

The next analysis projected group-averaged brain activation onto gradients (based on previous work) and used gradient timecourses to predict the behavioral report timecourses. This revealed that high activations in gradient 1 (sensory→association) predicted high sensory engagement, and that "episodic knowledge" thought patterns were predicted by increased visual cortex activations. Then, permutation tests were performed to see whether these thought pattern related activations corresponded to well defined regions on a given cluster.

In conclusion, this study tackles a highly interesting subject and does it creatively and expertly.

---

## [Author Response]

The following is the authors’ response to the previous reviews.

**Recommendations for the Authors:**

**Reviewer #2:**
(1) In my previous review, I noted that using three different movies to conclude that different genres evoke different thought patterns is an overinterpretation with only one instance per genre. In the rebuttal letter, the authors state that they provide "evidence that is necessary but not sufficient to conclude that we can distinguish different genres of films" (page 15). Accordingly, I suggest refraining from statements such as "There was a significant main effect of movie genre on memory" (page 13) in the manuscript.

Thank you for this point. We have removed any reference to genre.

Page 18 (referring to page 13) [354-355] “First, there was a significant main effect of movie on memory, F(2, 254.12) = 49.33, p <.001, η2 = .28.”

**Reviewer #3:**
The revised manuscript is easier to read and better contextualized.

Thank you for this comment and for your feedback to allow us to make the manuscript more clear.

**Public Reviews:**

**Reviewer #1:**
The lack of direct interrogation of individual differences/reliability of the mDES scores warrants some pause.

Our study's goal was to understand how group-level patterns of thought in one group of participants relate to brain activity in a different group of participants. To this end, we decomposed trial-level mDES data to show dimensions that are common across individuals, which demonstrated excellent split-half reliability. Then we used these data in two complementary ways. First, we established that these ratings reliably distinguished between the different films (showing that our approach is sensitive to manipulations of semantic and affective features in a film) and that these group-level patterns were also able to predict patterns of brain activity in a different group of participants (suggesting that mDES dimensions are also sensitive to the way brain activity emerges during movie watching). Second, we established that variation across individuals in their mDES scores predicted their comprehension of information from films. Thus our study establishes that when applied to movie-watching, mDES is sensitive to individual differences in the movie-watching experience (as determined by an individual's comprehension). Given the success of this study and the relative ease with which mDES can be performed, it will be possible in the future to conduct mDES studies that hone in on both the general features of the movie-watching experience, as well as aspects that are more unique to an individual.

**Reviewer #2:**
(1) The distinction between thinking and stimulus processing (in the sense of detecting and assigning meaning to features, modulated by factors such as attention) remains unclear. Is "thinking" a form of conscious access or a reportable read-out from sensory and higher-level stimulus processing? Or does it simply refer to the method used here to identify different processing states?

Thank you for highlighting this first point, which is an important consideration when attempting to map cognitive states. We have added some additional comments to our discussion section to expand on this point.

Page 35-36 [698-711] “It is possible, therefore, that the identification of regions of visual and auditory cortex by our study reflects the participants attention to sensory input, rather than the complex analysis of these inputs that may be required for certain features of the movie watching experience. On the other hand, it is possible that the movie-watching state is a qualitatively different type of mental state to those that emerge in typical task situations. For example, unlike tasks, the movie-watching state is characterized by multi-modal sensory input, semantically rich themes, that evolve together to reveal a continuous narrative to the viewer. It is possible, therefore, that movies engender an absorbed state which depends more on processing in sensory cortex than would occur in traditional task paradigms such as a working memory task (when systems in association cortex may be needed to maintain information related to task rules). Important headway into addressing this uncertainty can be achieved by using mDES to compare the types of states that occur in different contexts (including both movies and tasks) and comparing the topography of brain activity associated with different experiential states.”

(2) The dimensions of thought appear to be directly linked to brain areas traditionally associated with core faculties of perception and cognition. For example, superior temporal cortex codes for speech information, which is also where thought reports on verbal detail localize in this study. This raises the question of whether the present study truly captures mechanisms specific to thinking and distinct from processing, especially given that individual variations in reports were not considered and movie-specific features were not controlled for.

Thank you for this point, we have added an additional paragraph to the discussion to expand on this.

Page 35 [692-698] “Finally, it is worth considering whether the patterns of brain activity identified by our analysis reflect the stimuli that are processed during movie watching, or the cognitive and affective processing of this information. On the one hand, the regions we found were often within regions of sensory cortex, areas of the brain which are often ascribed basic stimulus processing functions [1]. Moreover, according to perspectives on cognition derived from more traditional task paradigms, complex features of cognition, such as the regulation of thought, are often attributed to regions of association cortex, such as the dorsolateral prefrontal cortex [2].”

**Reviewer #3:**
This paper is framed as presenting a new paradigm but it does little to discuss what this paradigm serves, what are its limitations and how it should have been tested. The novelty appears to be in using experience sampling from 1 sample to model the responses of a second sample.

Thank you for this comment, we have since made clear what the novelty of the methodology is, as you have correctly identified, by expanding this point beyond the methods section to clearly orient the reader to the application and limitation of our methodological approach with our paradigm.

Page 7-8 [149-174] “One challenge that arises when attempting to map the dynamics of thought onto brain activity during movie-watching is accounting for the inherently disruptive nature of experience sampling: to measure experience with sufficient frequency to map experiential reports during movies would inherently disrupt the natural processes of the brain and alter the viewer’s experience (for example, by pausing the film at a moment of suspense). Therefore, if we periodically interrupt viewers to acquire a description of their thoughts while recording brain activity, this could impact on the ability to capture important dynamic features of the brain. On the other hand, if we measured fMRI activity continuously over movie-watching (as is usually the case), we would lack the capacity to directly relate brain signals to the corresponding experiential states. Thus, to overcome these obstacles, we developed a novel methodological approach using two independent samples of participants. In the current study, one set of 120 participants was probed with mDES five times across the three ten-minute movie clips (11 minutes total, no sampling in the first minute). We used a jittered sampling technique where probes were delivered at different intervals across the film for different people depending on the condition they were assigned. Probe orders were also counterbalanced to minimize the systematic impact of prior and later probes at any given sampling moment. We used these data to construct a precise description of the dynamics of experience for every 15 seconds of three ten-minute movie clips. These data were then combined with fMRI data from a different sample of 44 participants who had already watched these clips without experience sampling [3]. By combining data from two different groups of participants, our method allows us to describe the time series of different experiential states (as defined by mDES) and relate these to the time series of brain activity in another set of participants who watched the same films with no interruptions. In this way, our study set out to explicitly understand how the patterns of thoughts that dominate different moments in a film in one group of participants relate to the brain activity at these time points in a second set of participants and, therefore, better understand the contribution of different neural systems to the movie-watching experience.”

Page 33-35 [658-691] “Importantly, our study provides a novel method for answering these questions and others regarding the brain basis of experiences during films that can be applied simply and cost-effectively. As we have shown, mDES can be combined with existing brain activity, allowing information about both brain activity and experience to be determined at a relatively low cost. For example, the cost-effective nature of our paradigm makes it an ideal way to explore the relationship between cognition and neural activity during movie-watching during different genres of film. In neuroimaging, conclusions are often made using one film in naturalistic paradigm studies [4]. Although the current study only used three movie clips, restraining our ability to form strong conclusions regarding how different patterns of thought relate to specific genres of film, in the future, it will be possible to map cognition across a more extensive set of movies and discern whether there are specific types of experience that different genres of films engage. One of the major strengths of our approach, therefore, is the ability to map thoughts across groups of participants across a wide range of movies at a relatively low cost.

Nonetheless, this paradigm is not without limitations. This is the first study, as far as we know, that attempts to compare experiential reports in one sample of participants with brain activity in a second set of participants, and while the utility of this method enables us to understand the relationship between thought and brain activity during movies, it will be important to extend our analysis to mDES data during movie-watching while brain activity is recorded. In addition, our study is correlational in nature, and in the future, it could be useful to generate a more mechanistic understanding of how brain activity maps onto the participants experience. Our analysis shows that mDES is able to discriminate between films, highlighting its broad sensitivity to variation in semantic or affective content. Armed with this knowledge, we propose that in the future, researchers could derive mechanistic insights into how the semantic features may influence the mDES data. For example, it may be possible to ask participants to watch movies in a scrambled order to understand how the structure of semantic or information influences the mapping between brains and ongoing experience as measured by mDES. Finally, our study focused on mapping group-level patterns of experience onto group-level descriptions of brain activity. In the future it may be possible to adopt a “precision-mapping” approach by measuring longer periods of experience using mDES and determining how the neural correlates of experience vary across individuals who watched the same movies while brain activity was collected [5]. In the future, we anticipate that the ease with which our method can be applied to different groups of individuals and different types of media will make it possible to build a more comprehensive and culturally inclusive understanding of the links between brain activity and movie-watching experience.”

What are the considerations for treating high-order thought patterns that occur during film viewing as stable enough to use across participants? What would be the limitations of this method? (Do all people reading this paper think comparable thoughts reading through the sections?) This is briefly discussed in the revised manuscript and generally treated as an opportunity rather than as a limitation.

It is likely, based on our study, that films can evoke both stereotyped thought patterns (i.e. thoughts that many people will share) and others that are individualistic. It is clear that, in principle, mDES is capable of capturing empirical information on both stereotypical thoughts and idiosyncratic thoughts. For example, clear differences in experiences across films and, in particular, during specific periods within a film, show that movie-watching can evoke broadly similar thought patterns in different groups of participants (see Figure 3 right-hand panel). On the other hand, the association between comprehension and the different mDES components indicate that certain individuals respond to the same film clip in different ways and that these differences are rooted in objective information (i.e. their memory of an event in a film clip). A clear example of these more idiosyncratic features of movie watching experience can be seen in the association between “Episodic Knowledge” and comprehension. We found that “Episodic Knowledge” was generally high in the romance clip from 500 Days of Summer but was especially high for individuals who performed the best, indicating they remembered the most information. Thus good comprehends responded to the 500 Days of Summer clip with responses that had more evidence of “Episodic Knowledge” In the future, since the mDES approach can account for both stereotyped and idiosyncratic features of experience, it will be an important tool in understanding the common and distinct features that movie watching experiences can have, especially given the cost effective manner with which these studies can be run.

In conclusion, this study tackles a highly interesting subject and does it creatively and expertly. It fails to discuss and establish the utility and appropriateness of its proposed method.

Thank you very much for your feedback and critique. In our revision and our responses to these questions, we provided more information about the method's robustness utility and application to understanding cognition. Thank you for bringing these points to our attention.

References

(1) Kaas, J.H. and C.E. Collins, *The organization of sensory cortex.* Current Opinion in Neurobiology, 2001. 11(4): p. 498-504.(2) Turnbull, A., et al., *Left dorsolateral prefrontal cortex supports context-dependent prioritisation of off-task thought.* Nature Communications, 2019. 10.(3) Aliko, S., et al., *A naturalistic neuroimaging database for understanding the brain using ecological stimuli.* Scientific Data, 2020. 7(1).(4) Yang, E., et al., *The default network dominates neural responses to evolving movie stories.* Nature Communications, 2023. 14(1): p. 4197.(5) Gordon, E.M., et al., *Precision Functional Mapping of Individual Human Brains.* Neuron, 2017. 95(4): p. 791-807.e7.